# Adaptive Regularization for Large-Scale Sparse Feature Embedding Models

**Mang Li** *, **Wei Lyu** *
Institute of Intelligent Technology
Alibaba International Digital Commerce Group
Hang Zhou, China
{mang.ll, lw386934}@alibaba-inc.com

## Abstract

The one-epoch overfitting problem has drawn widespread attention, especially in CTR and CVR estimation models in search, advertising, and recommendation domains. These models which rely heavily on large-scale sparse categorical features, often suffer a significant decline in performance when trained for multiple epochs. Although recent studies have proposed heuristic solutions, the fundamental cause of this phenomenon remains unclear. In this work, we present a theoretical explanation grounded in Rademacher complexity, supported by empirical experiments, to explain why overfitting occurs in models with large-scale sparse categorical features. Based on this analysis, we propose a regularization method that constrains the norm budget of embedding layers adaptively. Our approach not only prevents the severe performance degradation observed during multi-epoch training, but also improves model performance within a single epoch. This method has already been deployed in online production systems.

## 1 Introduction

Click-through rate (CTR) and conversion rate (CVR) estimation are critical for advertising, search and recommendation (ASR) applications. E-commerce platforms like Amazon and Taobao rely on optimizing CTR and CVR estimation to boost gross merchandise volume (GMV), while advertising platforms at Google and Meta depend on it to drive revenue growth. In the past decade, as deep learning has been widely adopted in ASR applications, most estimation models have been built on deep learning frameworks and rely on large-scale, sparse categorical features (Cheng et al. (2016); Guo et al. (2017); Lian et al. (2018)). Recent work (Zhang et al. (2022b)) demonstrates, through extensive experiments, that such models commonly suffer from the one-epoch overfitting phenomenon, where model performance drops sharply after the first epoch of training. It empirically suggests that the optimal practice is to train the models for only one epoch. At the same time, their empirical analysis also reveals a strong connection between feature sparsity and the one-epoch phenomenon. In ASR scenarios, categorical features can easily reach the scale of billions, and most feature values occur extremely infrequently (Xie et al. (2020); Zhang et al. (2022a)), which makes models especially susceptible to overfitting after the first epoch. Additionally, Ouyang et al. (2022) reported a similar one-epoch phenomenon during supervised fine-tuning (SFT) of large language model (LLM), though they argue that moderate overfitting can actually be beneficial. We leave it in future work. In this paper, we focus on addressing the multi-epoch overfitting problem for models with large-scale sparse features in ASR applications.

Liu et al. (2023) was the first to address the one-epoch problem using a heuristic approach. They introduced a multi-epoch data augmentation method (MEDA) that can be easily applied to estimation models with large-scale sparse categorical features. Later, they extended it to a continual learning paradigm (Fan et al. (2024)). To mitigate overfitting during multi-epoch training, MEDA reinitializes the embeddings of categorical features and their corresponding optimizer states at the beginning of each epoch. Although this approach effectively alleviates multi-epoch overfitting, it is

---

*Equal contribution.

fundamentally heuristic because the embedding parameters are only reinitialized at epoch boundaries. It neither ensures optimal convergence nor explains the root cause of overfitting. Moreover, this embedding reinitialization approach may result in the loss of substantial information, potentially resulting in suboptimal model performance. Nevertheless, it provides important inspiration for subsequent research. Another method, proposed by Wang et al. (2025), adopts an LLM-style paradigm. In this approach, generative pretraining is used to obtain embeddings for categorical features, which remain frozen during subsequent training. This strategy also helps to mitigate overfitting. However, it requires significant training resources to obtain the pretrained embeddings, and the performance cannot be fairly compared to single-epoch training because the parameter budget used during the pretraining phase is not included. Therefore, it cannot be regarded as a solution to the one-epoch problem, although it does highlight the pivotal role of the embedding layer in this phenomenon.

There are several well-established and general-purpose methods to mitigate overfitting, such as dropout (Srivastava et al. (2014)), $L_1$ and $L_2$ regularization (Schmidt et al. (2009); Moore & DeNero (2011)), and weight decay (Hanson & Pratt (1988)). However, in real-world industrial applications, data is typically high-dimensional and sparse, with different features exhibiting varying degrees of sparsity. Consequently, methods like AdamW (Loshchilov & Hutter (2017)), which apply the same weight decay across all parameters, tend to be suboptimal for addressing overfitting. Specifically, such approaches may reduce the fitting accuracy for relatively dense features while failing to effectively control overfitting in sparse features.

This paper presents a foundational theoretical analysis of the root cause of one-epoch overfitting, and proposes a method that adaptively determines regularization coefficients according to the effective update interval of each categorical feature value, and integrates these coefficients into the optimizers update rules. The proposed approach is particularly well-suited for models with large-scale sparse features. It reduces regularization for relatively dense features and neural network parameters, while assigning appropriate regularization strengths to individual categorical features. As a result, this method not only prevents the sharp performance degradation observed in multi-epoch training, but also enhances performance within a single epoch.

## 2 PRELIMINARY

In the industrial ASR domain, mainstream estimation models typically combine embedding layers with a variety of dense multi-layer perceptron (MLP) backbones, such as Gu et al. (2022). Taking an e-commerce platform estimation model as an example, these models process large-scale sparse inputs, including item IDs, brand IDs, and seller IDs, where the scale of categorical features can range from millions to billions. Before being processed by the estimation model, these sparse features are first mapped to low-dimensional dense representations via embedding layers. The one-hot categorical feature $\boldsymbol{x}_i(t) \in \mathbb{R}^{N_i}$ is defined as $\boldsymbol{x}_i(t) = [0, 0, \ldots, 1, \ldots, 0]^\top$, $i \in [S], t \in [T]$, which can be represented by an embedding vector $\boldsymbol{e}_i(t) \in \mathbb{R}^{d_i}$ shown below

$$\boldsymbol{e}_i(t) = \boldsymbol{E}_i^\top \boldsymbol{x}_i(t) \tag{1}$$

where $\boldsymbol{E}_i \in \mathbb{R}^{N_i \times d_i}$ is the embedding matrix for feature $i$ and $d_i$ is the embedding size. $T \in \mathbb{N}$ is the number of training samples. $S \in \mathbb{N}$ denotes the number of categorical features. $N_i \in \mathbb{N}, \forall i \in [S]$ denotes the number of distinct values for feature $i$. At each update step, an embedding lookup is performed to retrieve the dense representations for the sparse features via equation 1, which are then fed into the MLP layers. For simplicity, we separate the model into the embedding component and the MLP component denoted as $f$. In our theoretical analysis, we use a basic DNN backbone and provide experiments in section 4 to demonstrate the generalizability of our method with other backbone architectures.

### 2.1 NEURAL NETWORK DEFINITION

The general form of basic DNN function can be defined as

$$f(\cdot) = \boldsymbol{W}_L \sigma_{L-1} (\boldsymbol{W}_{L-1} \sigma_{L-2} (\ldots \sigma_1 (\boldsymbol{W}_1 \cdot))) \tag{2}$$

where $L$ denotes the number of MLP layers, $\sigma(\cdot)$ is the ReLU function (Agarap (2018)), $\boldsymbol{W}_l$ is the linear projection matrix of layer $l, l \in [L]$. In this paper, we only discuss the impact of embedding

layer so we can define the $t$-th sample output of DNN as

$$y(t) = f\left(\left[\boldsymbol{E}_1^\top \boldsymbol{x}_1(t); \boldsymbol{E}_2^\top \boldsymbol{x}_2(t); \dots; \boldsymbol{E}_S^\top \boldsymbol{x}_S(t)\right]\right) \tag{3}$$

where $y(t)$ is the logit output for CTR or CVR estimation model.

## 2.2 Rademacher Complexity Bound

Rademacher complexity is a standard tool to control the uniform convergence of given classes of predictors (Koltchinskii & Panchenko (2002),Zhang et al. (2016)). Formally, given a real-valued function class $\mathcal{H}$ with the input $\boldsymbol{z}$, we define the empirical Rademacher complexity $\widehat{\mathcal{R}}_T(\mathcal{H})$ as

$$\widehat{\mathcal{R}}_T(\mathcal{H}) = \mathbb{E}_{\boldsymbol{\epsilon}}\left[\sup_{h \in \mathcal{H}} \frac{1}{T} \sum_{t=1}^T \epsilon_t\, h(\boldsymbol{z}_t)\right] \tag{4}$$

where $\boldsymbol{\epsilon} = (\epsilon_1, \dots, \epsilon_T)$ is a vector whose entries $\epsilon_t$ are independent and uniformly distributed in $\{-1, +1\}$. Using standard arguments, such bounds, as long as the norm of $\boldsymbol{z}_t$ is bounded, can be converted to bounds on the generalization error, assuming access to a sample of $T$ i.i.d. training samples.

Based on Theorem 1 of Golowich et al. (2018), and noting that the embedding layer can be regarded as a linear projection, we can obtain the following upper bound on the Rademacher complexity of equation 3 as

$$\widehat{\mathcal{R}}_T(\mathcal{H}_L) \le \frac{1}{T}\left(\prod_{l=1}^L M_F(l)\right)\left(\sqrt{\sum_{i=1}^S M_{E_i}^2}\right)\left(\sqrt{2\log(2)L} + 1\right)\sqrt{\sum_{t=1}^T \sum_{i=1}^S \|\boldsymbol{x}_i(t)\|^2} \tag{5}$$

where $\|\cdot\|$ denotes $\ell_2$ norm, and each matrix $\boldsymbol{W}_l$ in function $f$ has Frobenius norm at most $M_F(l), l \in [L]$. $M_{E_i}$ is the Frobenius norm of embedding matrix $\boldsymbol{E}_i$ and the activation functions are assumed to be 1-Lipschitz and positive-homogeneous. Furthermore, because $\forall i, \|\boldsymbol{x}_i(t)\|^2 = 1$ this upper bound can be rewritten as

$$\widehat{\mathcal{R}}_T(\mathcal{H}_L) \le \sqrt{\frac{S}{T}}\left(\prod_{l=1}^L M_F(l)\right)\left(\sqrt{\sum_{i=1}^S \sum_{j=1}^{N_i} \tau_{ij}}\right)\left(\sqrt{2\log(2)L} + 1\right) \tag{6}$$

where $\tau_{ij}$ is the squared $\ell_2$ norm of $j$-th row of embedding matrix $\boldsymbol{E}_i$. In ASR applications, the majority of parameters are concentrated in the embedding layers. Thus, we can see that $\sum_{i=1}^S \sum_{j=1}^{N_i} \tau_{ij}$ has a significant impact on the upper bound of Rademacher complexity, and consequently, on the generalization error bound. In appendix D, we provide an upper bound on the Rademacher complexity for an FM-like model, which indicates that the embedding layers also have a significantly impact on the bound.

## 3 The Proposed Approach

As discussed in section 2, high-dimensional embedding matrices lead to increased generalization error bounds if the training loss remains constant. On the other hand, strictly constraining the norms of the embedding vectors to reduce Rademacher complexity may increase the training error, thus potentially degrading overall performance. To identify the optimal regularization factor, we formulate this trade-off as a constrained optimization problem, described as follows.

$$\min_{\tau_{ij} > 0} \sum_{i=1}^S \sum_{j=1}^{N_i} m_{ij}\varphi(\tau_{ij}) \quad \text{s.t.} \quad \sum_{i=1}^S \sum_{j=1}^{N_i} \tau_{ij} \le C \tag{7}$$

where $\varphi(\tau_{ij}) = \min_{\|\boldsymbol{e}_{ij}\|^2 \le \tau_{ij}} \mathcal{L}(\boldsymbol{e}_{ij})$ with $i \in [S]$ and $j \in [N_i]$. Here, $\boldsymbol{e}_{ij}$ denotes the $j$-th row of embedding matrix $\boldsymbol{E}_i$, and $\mathcal{L}(\boldsymbol{e}_{ij})$ represents the average cross-entropy (CE) loss evaluated on the DNN output when the training sample activates $\boldsymbol{e}_{ij}$. For the CE loss function, $\mathcal{L}(\boldsymbol{e}_{ij})$ is lower semi-continuous and bounded below on bounded sets (Goodfellow et al. (2016)). Consequently, $\varphi(\tau_{ij})$

is well-defined (Ok (2011)) and monotonically non-increasing in $\tau_{ij}$, since the feasible set expands with $\tau_{ij}$. Here, $\tau_{ij}$ is the squared norm budget of embedding vector $e_{ij}$. According to DiBenedetto (2016), $\varphi(\tau_{ij})$ is differentiable almost everywhere, and we assume that $\varphi(\tau_{ij})$ is differentiable[1] at the optimal points $\tau_{ij}^*$ for analysis. The coefficient $m_{ij}$ denotes the sample frequency with which embedding vector $e_{ij}$ appears in the training dataset and $C$ is a constant that defines the global norm upper bound for embedding layers. As shown in equation 6, the value of $C$ directly determines the Rademacher complexity upper bound.

**Proposition 1** *A necessary condition for the optimal regularization multiplier $\lambda_{ij}^*$ associated with the $\|e_{ij}\|^2 \leq \tau_{ij}^*$ is given by $\lambda_{ij}^* = \mu_0/m_{ij}$, where $\mu_0$ is the Lagrange multiplier corresponding to $\sum_{i=1}^{S} \sum_{j=1}^{N_i} \tau_{ij}^* \leq C$.*

*Proof.* Based on the above assumption, at points of differentiability where a standard constraint qualification holds for the inner optimal solution, the envelope theorem and Lagrangian decomposition (Shapiro (1979)) yield

$$\varphi'(\tau_{ij}^*) = -\lambda_{ij}^* \tag{8}$$

where $\lambda_{ij}^* \geq 0$ is the Lagrange multiplier corresponding to $\|e_{ij}\|^2 \leq \tau_{ij}^*$, and $\tau_{ij}^*$ is the optimal solution of equation 7. Here, $\varphi'(\tau_{ij}^*)$ denotes the derivative of $\varphi(\tau_{ij}^*)$ with respect to $\tau_{ij}^*$. We reformulate the optimization problem defined in equation 7 with the Lagrange multipliers $\mu_{ij}$ and $\mu_0$, corresponding to the nonnegativity constraints of $\tau_{ij} \geq 0$ and $\sum_{i=1}^{S} \sum_{j=1}^{N_i} \tau_{ij} \leq C$ respectively, where $\mu_{ij}$ and $\mu_0$ are restricted to be non-negative.

$$\min_{\tau_{ij} > 0} \left( \sum_{i=1}^{S} \sum_{j=1}^{N_i} m_{ij}\varphi(\tau_{ij}) + \mu_0 \left( \sum_{i=1}^{S} \sum_{j=1}^{N_i} \tau_{ij} - C \right) - \sum_{i=1}^{S} \sum_{j=1}^{N_i} \mu_{ij}\tau_{ij} \right) \tag{9}$$

$$\text{s.t. } \mu_0 \geq 0, \mu_{ij} \geq 0, \forall i \in [S], \forall j \in [N_i]$$

Based on the KKT condition (Boyd & Vandenberghe (2004)), we have $m_{ij}\varphi'(\tau_{ij}^*) + \mu_0 - \mu_{ij} = 0$. Applying the complementary slackness condition, we obtain $\mu_{ij}\tau_{ij}^* = 0$. This implies the optimality condition for $\tau_{ij} > 0$ can be simplified as

$$m_{ij}\varphi'(\tau_{ij}^*) + \mu_0 = 0 \tag{10}$$

By substituting equation 8 into equation 10, we obtain a necessary condition for the optimal regularization multiplier $\lambda_{ij}^* = \mu_0/m_{ij}$.

## 3.1 ADAPTIVE REGULARIZATION METHOD

As shown in equation 6, embedding layers in ASR applications typically take the majority of the model parameters and have a substantial impact on the generalization error bound. Proposition 1 suggests that the norm budget for each embedding vector should be allocated according to its sample frequency. However, it is not easy to use the frequency directly during the training process. We can estimate $m_{ij}$ via the stochastic occurrence interval $\mathcal{I}_{ij}$ of $e_{ij}$ for $i \in [S]$ and $j \in [N_i]$. Specifically, given that $x_i(t)$ is sampled from an i.i.d. distribution (a common assumption in deep learning), we have $\mathbb{E}[m_{ij}] = T/\mathbb{E}[\mathcal{I}_{ij}]$ (Papoulis (1965)). It enables us to incorporate frequency estimation into the norm budget allocation strategy during training.

Based on the analysis above, we propose an adaptive method that assigns the regularization strength based on the occurrence interval of each embedding vector. Specifically, at each training step $k$, we define last valid update step (LVS) for the embedding vector $e_{ij}$ as $s_{ij}^k$. If the gradient norm of $e_{ij}$ satisfies $\|g_{ij}\| > 0$, we update the LVS by setting $s_{ij}^k = k$. Otherwise, $s_{ij}^k$ retains its previous value. Therefore, $s_{ij}^k$ serves as a lazy-update variable. We define the update interval of step $k$ as $I_{ij}^k = k - s_{ij}^{k-1} - 1$. The adaptive regularization coefficient $\lambda_{ij}^k$ is then dynamically computed as below

$$\lambda_{ij}^k = \min\left(1, \alpha I_{ij}^k\right), i \in [S], j \in [N_i] \tag{11}$$

---

[1]In appendix C, we give a discussion for the non-smooth case.

Here, $\alpha \in [0,1)$ denotes the base regularization coefficient. Following the decoupled weight decay approach in AdamW (Loshchilov & Hutter (2017)), we incorporate the dynamically computed regularization into each optimizer update step. Algorithm 1 outlines the procedure of Adam with adaptive regularization (AdamAR).

For clarity and practical implementation, we use $\theta_p$ to denote parameters in the estimation model, where $p \in [P]$ and $P$ is the total number of parameters. At each update step $k$, we first compute the adaptive regularization $\lambda_p^k$ according to equation 11 and use it to update the corresponding model parameter $\theta_p^k$. We then identify the parameters whose gradient norm $\|g_p^k\|$ is greater than zero in current step and update their last update step state $s_p^k$ for use in the next iteration.

In fact, our method is not only suitable for Adam, but also compatible with various gradient-based optimizers which have weight decay factor, such as Adagrad (Duchi et al. (2011)). We have implemented an adaptive version for Adagrad in algorithm 2 of the appendix G. In section 4, we conducted comparative experiments evaluating performance across both Adam and Adagrad.

Moreover, the AdamAR algorithm requires additional storage to record the last valid update step. Further details of the computation and memory analysis are provided in appendix F.

---

**Algorithm 1** Adam with Adaptive Regularization (AdamAR)

---

1: given $\beta_1 = 0.9$, $\beta_2 = 0.999$, $\varepsilon = 10^{-8}$, $\alpha$, learning rate $\eta$
2: initialize time step $k \leftarrow 0$, parameter $\theta_p^{k=0}$, first moment $m_p^{k=0} \leftarrow 0$, second moment $v_p^{k=0} \leftarrow 0$, last update step state $s_p^{k=0} \leftarrow 0$
3: **repeat**
4:    $k \leftarrow k + 1$
5:    $g_p^k \leftarrow \nabla_{\theta_p} f\left(\theta_p^{k-1}\right)$ (Get gradients w.r.t. stochastic objective at timestep t)
6:    $m_p^k \leftarrow \beta_1 m_p^{k-1} + (1 - \beta_1) g_p^k$ (Update biased first moment estimate)
7:    $v_p^k \leftarrow \beta_2 v_p^{k-1} + (1 - \beta_2) \left(g_p^k\right)^2$ (Update biased second raw moment estimate)
8:    $\hat{m}_p^k \leftarrow m_p^k / \left(1 - \beta_1^k\right)$ (Compute bias-corrected first moment estimate)
9:    $\hat{v}_p^k \leftarrow v_p^k / \left(1 - \beta_2^k\right)$ (Compute bias-corrected second raw moment estimate)
10:   $\lambda_p^k \leftarrow \min\left(1, \left(k - s_p^{k-1} - 1\right)\alpha\right)$ (Compute adaptive regularization through equation 11)
11:   $s_p^k \leftarrow k$ if $\|g_p^k\| > 0$ else $s_p^{k-1}$ (Update the last update step when the gradient norm is greater than zero)
12:   $\theta_p^k \leftarrow \theta_p^{k-1} - \lambda_p^k \theta_p^{k-1} - \eta \cdot \hat{m}_p^k / \left(\sqrt{\hat{v}_p^k} + \varepsilon\right)$ (Update parameters)
13: **until** stopping criterion is met
14: return optimized parameters $\theta_p^t$

---

## 3.2 DISCUSSION ON THE MECHANISM OF REGULARIZATION

As shown in equation 7, if we do not constrain the embedding norm, the global norm will continue to grow until further increases no longer yield improvements in the objective value since $\varphi(\tau_{ij})$ is non-increasing. In other words, the embedding norms will continue to grow during training, resulting in a looser upper bound on the Rademacher complexity. The sharper drop in performance during multi-epoch training can be attributable to the increased Rademacher complexity resulting from unconstrained norm growth as demonstrated in the experimental section 4.4.

Then we discuss how the adaptive regularization takes effect, and also provides guidance for selecting the regularization coefficient $\alpha$.

**Proposition 2** *When adaptive regularization is applied according to equation 11, the update rule for parameters satisfy* $\|\theta_p^k\| \leq (1 - \alpha)^{I_p^k} \|\theta_p^{k-1}\| + \|\eta \cdot \hat{m}_p^k / (\sqrt{\hat{v}_p^k} + \varepsilon)\|$

*Proof.* Let $I_p^k \in \mathbb{Z}_{\geq 0} \cap [0, 1/\alpha)$, and $\alpha$ is typically chosen such that $\alpha \in [0, 1)$. By applying Bernoulli's inequality to $(1 - \alpha)^{I_p^k}$, we obtain $(1 - \alpha)^{I_p^k} \geq 1 - \alpha I_p^k$. In the case where $I_p^k \in \mathbb{Z}_{\geq 0} \cap [1/\alpha, \infty)$, it follows that $(1 - \alpha)^{I_p^k} > 0$ since $\alpha < 1$ and $I_p^k \geq 0$. Combining both cases, we

have

$$(1-\alpha)^{I_p^k} \geq 1 - \min\left(1, \alpha I_p^k\right) \tag{12}$$

We substitute equation 12 into $\theta_p^{k-1} - \lambda_p^k \theta_p^{k-1} - \eta \cdot \hat{m}_p^k / \left(\sqrt{\hat{v}_p^k} + \varepsilon\right)$, then we can have

$$\|\theta_p^k\| \leq (1-\alpha)^{I_p^k} \|\theta_p^{k-1}\| + \|\eta \cdot \hat{m}_p^k / (\sqrt{\hat{v}_p^k} + \varepsilon)\| \tag{13}$$

From proposition 2, the update rule for $\theta_p^k$ can be interpreted intuitively. If the interval $I_p^k$ is large for the corresponding sparse categorical features, the term $(1-\alpha)^{I_p^k} \|\theta_p^{k-1}\|$ becomes negligible, and $\theta_p^k$ is effectively determined by the latest gradient. In the ASR domain, certain categorical feature values may not appear in every mini-batch, resulting in their embedding parameters being updated infrequently, while the MLP parameters are updated in every batch. After the MLP component has nearly converged, the embedding parameters associated with sparse features will have received far fewer updates and may become misaligned with the current state of the MLP parameters, making the previous value $\theta_p^{k-1}$ less informative. Proposition 2 demonstrates that the factor $(1-\alpha)^{I_p^k}$ exponentially attenuates the previous value $\theta_p^{k-1}$, making the embedding parameters of sparse features depend more heavily on the current gradient. Meanwhile, since the update interval $I_p^k$ for MLP parameters is identically zero (as they are updated in every batch), the regularization primarily affects the embedding parameters corresponding to very low-frequency feature values. Moreover, we observe that the method proposed in Liu et al. (2023) is a special case of our approach when zero reinitialization is applied to the embedding and $I_p^k$ is specified as

$$I_p^k = \begin{cases} 1/\alpha, & \text{if } kB \bmod T = 0 \\ 0, & \text{otherwise} \end{cases} \tag{14}$$

Where $B$ is the batch size. It can be seen that this heuristic is primarily effective at epoch boundaries. However, for features that have already become overfitted within a single epoch, its performance remains suboptimal.

### 3.3 MINIMUM CONVERGENCE

We analyze the convergence of adaptive regularization methods in the non-convex setting using the minimum convergence rules proposed by Khaled & Richtárik (2020). The assumptions underlying our convergence analysis are listed below.

**Assumptions 1**. The function $f$ is differentiable and its gradient is Lipschitz continuous, i.e., there exists $L_0 > 0$ such that $\|\nabla f(\boldsymbol{\theta}^{k+1}) - \nabla f(\boldsymbol{\theta}^k)\| \leq L_0 \|\boldsymbol{\theta}^{k+1} - \boldsymbol{\theta}^k\|, \forall k \geq 1$, and $f$ is lower bounded at the optimal solution, i.e., $f^* > -\infty$.

**Assumptions 2**. $\boldsymbol{g}^k$ is an unbiased estimator of the full gradient, i.e., $\mathbb{E}\left[\boldsymbol{g}^k\right] = \nabla f(\boldsymbol{\theta}^k)$ with $M > 0$, and the algorithm accesses a bounded stochastic gradient, i.e., $\|\boldsymbol{g}^k\| \leq M$ a.s.

**Proposition 3** *The adaptive regularization method preserves the minimum convergence bound of the Adam optimizer with stochastic conditions, which can be expressed as*

$$\min_{1 \leq k \leq K} \mathbb{E}\left[\left\|\nabla f\left(\boldsymbol{\theta}^k\right)\right\|^2\right] \leq \frac{C_1 + C_2 \sum_{k=1}^K \eta_k + C_3 \sum_{k=1}^K \eta_k^2}{\sum_{k=1}^K \eta_k} \tag{15}$$

*where $C_1$, $C_2$ and $C_3$ are constants, $\eta_k$ denotes the step size at iteration $k$ out of $K$ total iterations.*

The proof, along with the definitions of the constants $C_1$, $C_2$, and $C_3$, can be found in appendix B.1. Proposition 3 shows that neither the random noise nor the deterministic regularization parameter affects the minimum convergence of Adam, and it only changes the constant term in equation 15.

## 4 EXPERIMENTAL VALIDATION

To evaluate the effectiveness of our proposed method, we conduct several experiments on different public datasets and our industrial dataset. The experimental setup is described in section 4.1. The

training framework proposed by Zhu et al. (2022) is used for training on public datasets, while XDL (Jiang et al. (2019)) is employed for training our proprietary industrial dataset. We show the learning curve on the Avazu dataset in section 4.2 to demonstrate the better generalization of our methods over multi-epoch training. In section 4.3, we compare multiple backbones and datasets to prove the generalization of our method. We give a detailed example to show that the low-frequency embeddings are the primary contributors of the one-epoch problem in section 4.4, which demonstrates we should allocate a smaller norm budget to embeddings with lower sample frequency. Finally, in section 4.5, we conduct an ablation study to further clarify the contribution of occurrence interval estimation. For reproduction, the code[2] is available, and we use the same seed for identical experimental settings.

## 4.1 EXPERIMENTAL SETUP

**Datasets**. In our experiments, we use three public datasets iPinYou[3], Amazon[4], Avazu[5] along with LZD, a proprietary online business dataset from sponsored search. Dataset details are provided in appendix L.

**MLP Backbones**. We evaluate our method using four MLP backbones, namely DNN, WDL (Cheng et al. (2016)), xDeepFM (Lian et al. (2018)) and WuKong (Zhang et al. (2024))

**Methods**. To evaluate the one-epoch overfitting phenomenon, we train the models for 4 epochs and compare 4 benchmarks. 1) Baseline optimizer: Adam and Adagrad serve as the baseline optimizers. 2) MEDA: MEDA is applied using the same optimizer settings as in baseline. 3) AdamW and AdagradW: The baseline optimizers are enhanced with weight decay only on embedding layers. 4) SAM: The baseline optimizers are combined with SAM (Foret et al. (2020)). 5) AdamAR and AdagradAR: The baseline optimizers are combined with our method.

**Hyperparameters**. The embedding dimension is set to 32, with zero initialization. The learning rate is set to 0.001 for Adam and 0.01 for Adagrad, respectively. The batch size is 2048. Both $\alpha$ and the weight decay parameter are selected via grid search over values of the form $10^n$ where $n$ ranges from $-6.5$ to $-1$ with step size of 0.5. The optimal hyperparameters are selected based on validation performance (see appendix I for details). Network architectures are configured with minor adjustments across datasets. For detailed configurations, please refer to table 6 in appendix H. All other hyperparameters are set to their default values. Area under the curve (AUC) and binary cross-entropy loss are used as evaluation metrics. Experiments are conducted on a single machine with an NVIDIA L20 GPU.

## 4.2 LEARNING CURVE AND GENERALIZATION RESULTS

In this section, we present the training loss and test AUC across multiple epochs on the Avazu dataset, using a DNN backbone and the Adam optimizer. The learning curves in figure 1 compare our method with several baselines. For the native optimizer, training loss decreases rapidly after the first epoch. However, test AUC drops sharply with additional epochs, indicating clear overfitting. In contrast, MEDA and weight decay alleviate this problem, yielding more stable test AUC throughout training. Our method achieves the best overall performance. Figures 1(b) and 1(c) reveal an inverse correlation between the $\ell_2$ norm of the embedding vectors and test AUC. It demonstrates that constraining the embedding norm improves generalization ability of models. Among all methods, our proposed AdamAR achieves the lowest cumulative $\ell_2$ norm for the embedding vectors, highlighting its effectiveness in controlling regularization strength.

## 4.3 PERFORMANCE OVER DIFFERENT DATASETS AND MLP BACKBONES

In this section, we compare test AUC across different datasets and MLP backbones to demonstrate the generalization capability of our method. Each experiment is repeated three times with different seed. We report the average test AUC using Adam and Adagrad optimizers in table 1 and table

---

[2]https://github.com/alibaba-aidc/adaptive-regularization.git
[3]https://huggingface.co/datasets/reczoo/iPinYou_x1
[4]https://huggingface.co/datasets/reczoo/AmazonElectronics_x1
[5]https://huggingface.co/datasets/reczoo/Avazu_x1

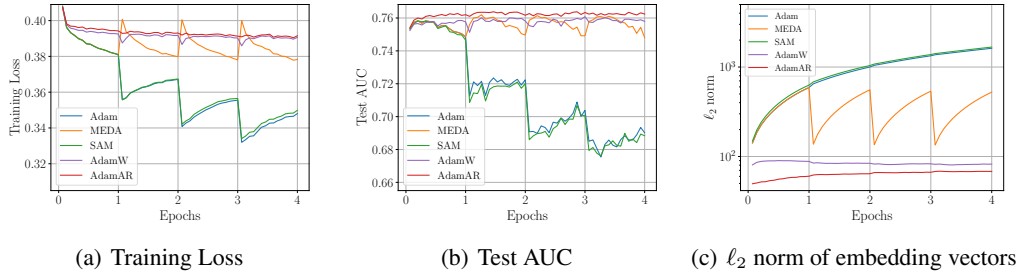

| (a) Training Loss | (b) Test AUC | (c) $\ell_2$ norm of embedding vectors |

Figure 1: Performance of four methods on Avazu dataset with DNN backbone. (a) shows the training loss curves. (b) presents the test AUC. (c) illustrates the cumulative $\ell_2$ norm of embedding vectors.

2, while the standard deviation and detailed scalability comparison are presented in appendix J. The results show that our methods consistently outperform MEDA and weight decay method on all datasets and architectures in single-epoch training, except for the Amazon dataset which has less features and samples where SAM achieves superior performance. Notably, in multi-epoch training, our method achieves the highest AUC, surpassing all other approaches across every dataset and architecture. Furthermore, the gains are consistent across diverse model architectures, ranging from basic DNN to more sophisticated designs like WuKong, which capture complex feature interactions. Overall, these results demonstrate the robustness and versatility of our adaptive regularization approach across various settings.

Table 1: Comparison of average test AUC across different datasets and models using Adam optimizer. E1-E4 denote results after 1-4 epochs, respectively. The best results are highlighted in bold.

| Dataset | Method | DNN | | | | WDL | | | | xDeepFM | | | | WuKong | | | |
|---|---|---|---|---|---|---|---|---|---|---|---|---|---|---|---|---|---|
| | | E1 | E2 | E3 | E4 | E1 | E2 | E3 | E4 | E1 | E2 | E3 | E4 | E1 | E2 | E3 | E4 |
| iPinYou | Adam | 0.7515 | 0.7304 | 0.7061 | 0.7014 | 0.7619 | 0.7320 | 0.7028 | 0.6987 | 0.7590 | 0.7391 | 0.6969 | 0.6844 | 0.7611 | 0.7442 | 0.7082 | 0.6915 |
| | MEDA | 0.7515 | 0.7644 | 0.7684 | 0.7717 | 0.7619 | 0.7589 | 0.7565 | 0.7551 | 0.7590 | 0.7584 | 0.7575 | 0.7597 | 0.7611 | 0.7663 | 0.7662 | 0.7706 |
| | SAM | 0.7510 | 0.7593 | 0.7445 | 0.7256 | 0.7610 | 0.7581 | 0.7404 | 0.7248 | 0.7565 | 0.7517 | 0.7413 | 0.7274 | 0.7033 | 0.7485 | 0.7465 | 0.7306 |
| | AdamW | 0.7475 | 0.7592 | 0.7623 | 0.7568 | 0.7551 | 0.7656 | 0.7646 | 0.7634 | 0.7607 | 0.7660 | 0.7651 | 0.7574 | 0.7579 | 0.7634 | 0.7605 | 0.7511 |
| | AdamAR | **0.7566** | **0.7692** | **0.7688** | **0.7724** | **0.7655** | **0.7729** | **0.7670** | **0.7668** | **0.7725** | **0.7733** | **0.7711** | **0.7678** | **0.7653** | **0.7736** | **0.7748** | **0.7736** |
| Amazon | Adam | 0.8482 | 0.8548 | 0.8335 | 0.8180 | 0.8474 | 0.8510 | 0.8261 | 0.8156 | 0.8460 | 0.8535 | 0.8287 | 0.8163 | 0.8580 | 0.8600 | 0.8348 | 0.8232 |
| | MEDA | 0.8482 | 0.8544 | 0.8556 | 0.8573 | 0.8474 | 0.8506 | 0.8566 | 0.8566 | 0.8460 | 0.8519 | 0.8551 | 0.8562 | 0.8580 | 0.8611 | 0.8621 | 0.8626 |
| | SAM | 0.8507 | 0.8587 | 0.8417 | 0.8249 | **0.8516** | 0.8567 | 0.8396 | 0.8213 | **0.8505** | 0.8587 | 0.8390 | 0.8232 | **0.8588** | 0.8639 | 0.8404 | 0.8225 |
| | AdamW | 0.8476 | 0.8571 | 0.8426 | 0.8276 | 0.8461 | 0.8533 | 0.8380 | 0.8223 | 0.8446 | 0.8557 | 0.8381 | 0.8240 | 0.8564 | 0.8632 | 0.8478 | 0.8349 |
| | AdamAR | **0.8507** | **0.8683** | **0.8708** | **0.8686** | 0.8496 | **0.8654** | **0.8689** | **0.8659** | 0.8483 | **0.8676** | **0.8687** | **0.8675** | 0.8582 | **0.8696** | **0.8693** | **0.8664** |
| Avazu | Adam | 0.7461 | 0.7205 | 0.7014 | 0.6883 | 0.7483 | 0.7221 | 0.6982 | 0.6886 | 0.7488 | 0.7217 | 0.7019 | 0.6869 | 0.7514 | 0.7360 | 0.7141 | 0.7079 |
| | MEDA | 0.7461 | 0.7498 | 0.7489 | 0.7485 | 0.7483 | 0.7488 | 0.7480 | 0.7494 | 0.7488 | 0.7489 | 0.7505 | 0.7506 | 0.7514 | 0.7548 | 0.7553 | 0.7571 |
| | SAM | 0.7451 | 0.7194 | 0.7013 | 0.6899 | 0.7477 | 0.7205 | 0.6993 | 0.6902 | 0.7484 | 0.7190 | 0.7010 | 0.6902 | 0.7513 | 0.7333 | 0.7155 | 0.7156 |
| | AdamW | 0.7572 | 0.7582 | 0.7582 | 0.7583 | 0.7585 | 0.7581 | 0.7563 | 0.7570 | 0.7583 | 0.7582 | 0.7589 | 0.7593 | 0.7542 | 0.7547 | 0.7558 | 0.7564 |
| | AdamAR | **0.7617** | **0.7631** | **0.7629** | **0.7629** | **0.7629** | **0.7629** | **0.7627** | **0.7626** | **0.7628** | **0.7633** | **0.7638** | **0.7636** | **0.7624** | **0.7612** | **0.7623** | **0.7624** |
| LZD | Adam | 0.7118 | 0.6613 | 0.6252 | 0.6065 | 0.7155 | 0.6726 | 0.6308 | 0.6079 | 0.7164 | 0.6787 | 0.6321 | 0.6050 | 0.7101 | 0.6645 | 0.6102 | 0.6044 |
| | MEDA | 0.7118 | 0.7105 | 0.7081 | 0.7176 | 0.7155 | 0.7162 | 0.7177 | 0.7162 | 0.7164 | 0.7170 | 0.7152 | 0.7139 | 0.7101 | 0.7170 | 0.7129 | 0.7128 |
| | SAM | 0.7130 | 0.6696 | 0.6341 | 0.6155 | 0.7161 | 0.6795 | 0.6360 | 0.6111 | 0.7166 | 0.6750 | 0.6344 | 0.6134 | 0.7146 | 0.6713 | 0.6443 | 0.6212 |
| | AdamW | 0.7132 | 0.7135 | 0.7140 | 0.7139 | 0.7143 | 0.7138 | 0.7142 | 0.7131 | 0.7142 | 0.7151 | 0.7139 | 0.7139 | 0.7115 | 0.7135 | 0.7152 | 0.7142 |
| | AdamAR | **0.7229** | **0.7235** | **0.7241** | **0.7234** | **0.7233** | **0.7240** | **0.7246** | **0.7240** | **0.7244** | **0.7256** | **0.7242** | **0.7238** | **0.7227** | **0.7215** | **0.7208** | **0.7202** |

## 4.4 Example of the Root Cause of Overfitting

In this section, we give an example to show the root cause of multi-epoch overfitting. We use the iPinYou dataset with DNN backbone and Adam to illustrate this issue by single experiment. The detailed feature statistics are listed in table 7 in appendix K. We can observe that most features on iPinYou dataset are relatively dense. To investigate the root cause of overfitting, we select the feature "IP", which is the most sparse feature in iPinYou dataset with 704,966 unique IDs. We apply a filtering procedure to reduce its feature sparsity. Given a ratio $r$, we only retain the top-$r$ fraction (by frequency) of IDs and replace other IDs with a default ID.

Table 2: Comparison of average test AUC across different datasets and models using Adagrad optimizer. E1-E4 denote results after 1-4 epochs, respectively. The best results are highlighted in bold.

| Dataset | Method | DNN | | | | WDL | | | | xDeepFM | | | | WuKong | | | |
|---|---|---|---|---|---|---|---|---|---|---|---|---|---|---|---|---|---|
| | | E1 | E2 | E3 | E4 | E1 | E2 | E3 | E4 | E1 | E2 | E3 | E4 | E1 | E2 | E3 | E4 |
| iPinYou | Adagrad | 0.7593 | 0.6507 | 0.6231 | 0.6095 | 0.7646 | 0.6653 | 0.6321 | 0.6241 | 0.7674 | 0.6843 | 0.6505 | 0.6527 | 0.7661 | 0.6900 | 0.6497 | 0.6320 |
| | MEDA | 0.7593 | 0.7686 | 0.7710 | 0.7729 | 0.7646 | 0.7681 | 0.7685 | 0.7715 | 0.7674 | 0.7722 | 0.7728 | 0.7740 | 0.7661 | 0.7705 | 0.7729 | 0.7695 |
| | SAM | 0.7576 | 0.7661 | 0.7487 | 0.7303 | 0.7624 | 0.7676 | 0.7518 | 0.7369 | 0.7637 | 0.7661 | 0.7499 | 0.7363 | 0.7409 | 0.7678 | 0.7525 | 0.7427 |
| | AdagradW | 0.7558 | 0.7651 | 0.7667 | 0.7595 | 0.7593 | 0.7675 | 0.7635 | 0.7593 | 0.7665 | 0.7673 | 0.7666 | 0.7652 | 0.7578 | 0.7661 | 0.7649 | 0.7600 |
| | AdagradAR | **0.7681** | **0.7754** | **0.7760** | **0.7744** | **0.7731** | **0.7772** | **0.7748** | **0.7720** | **0.7760** | **0.7768** | **0.7762** | **0.7745** | **0.7718** | **0.7776** | **0.7782** | **0.7774** |
| Amazon | Adagrad | 0.8438 | 0.8402 | 0.8141 | 0.8042 | 0.8406 | 0.8345 | 0.8085 | 0.7982 | 0.8405 | 0.8374 | 0.8103 | 0.7996 | 0.8491 | 0.8472 | 0.8156 | 0.8046 |
| | MEDA | 0.8438 | 0.8481 | 0.8495 | 0.8505 | 0.8406 | 0.8470 | 0.8497 | 0.8510 | 0.8405 | 0.8463 | 0.8476 | 0.8500 | 0.8491 | 0.8569 | 0.8587 | 0.8609 |
| | SAM | **0.8500** | 0.8527 | 0.8361 | 0.8193 | **0.8490** | 0.7995 | 0.8325 | 0.8155 | **0.8483** | 0.8521 | 0.8325 | 0.8170 | **0.8556** | 0.8567 | 0.8356 | 0.8174 |
| | AdagradW | 0.8428 | 0.8578 | 0.8563 | 0.8535 | 0.8424 | 0.8553 | 0.8531 | 0.8498 | 0.8410 | 0.8565 | 0.8522 | 0.8508 | 0.8513 | 0.8630 | 0.8617 | 0.8606 |
| | AdagradAR | 0.8479 | **0.8659** | **0.8711** | **0.8712** | 0.8453 | **0.8631** | **0.8687** | **0.8707** | 0.8444 | **0.8641** | **0.8681** | **0.8703** | 0.8538 | **0.8690** | **0.8708** | **0.8700** |
| Avazu | Adagrad | 0.7541 | 0.7323 | 0.7164 | 0.7074 | 0.7543 | 0.7305 | 0.7158 | 0.7073 | 0.7550 | 0.7319 | 0.7160 | 0.7069 | 0.7548 | 0.7311 | 0.7160 | 0.7041 |
| | MEDA | 0.7541 | 0.7549 | 0.7544 | 0.7545 | 0.7543 | 0.7547 | 0.7540 | 0.7551 | 0.7550 | 0.7538 | 0.7550 | 0.7553 | 0.7548 | 0.7551 | 0.7556 | 0.7564 |
| | SAM | 0.7553 | 0.7333 | 0.7199 | 0.7094 | 0.7557 | 0.7328 | 0.7178 | 0.7079 | 0.7564 | 0.7332 | 0.7198 | 0.7094 | 0.7558 | 0.7353 | 0.7211 | 0.7110 |
| | AdagradW | 0.7578 | 0.7580 | 0.7575 | 0.7566 | 0.7584 | 0.7581 | 0.7567 | 0.7585 | 0.7583 | 0.7581 | 0.7580 | 0.7585 | 0.7524 | 0.7555 | 0.7555 | 0.7563 |
| | AdagradAR | **0.7628** | **0.7629** | **0.7630** | **0.7624** | **0.7631** | **0.7634** | **0.7623** | **0.7626** | **0.7636** | **0.7633** | **0.7635** | **0.7633** | **0.7628** | **0.7626** | **0.7627** | **0.7620** |
| LZD | Adagrad | 0.7226 | 0.6658 | 0.6433 | 0.6376 | 0.7244 | 0.6695 | 0.6389 | 0.6386 | 0.7222 | 0.6675 | 0.6400 | 0.6382 | 0.7247 | 0.6726 | 0.6412 | 0.6339 |
| | MEDA | 0.7226 | 0.7183 | 0.7150 | 0.7236 | 0.7244 | 0.7241 | 0.7239 | 0.7219 | 0.7222 | 0.7237 | 0.7253 | 0.7256 | 0.7247 | 0.7237 | 0.7234 | 0.7239 |
| | SAM | 0.7213 | 0.6761 | 0.6446 | 0.6222 | 0.7229 | 0.6839 | 0.6481 | 0.6266 | 0.7229 | 0.6854 | 0.6480 | 0.6268 | 0.7248 | 0.6802 | 0.6479 | 0.6299 |
| | AdagradW | 0.7145 | 0.7132 | 0.7135 | 0.7154 | 0.7138 | 0.7149 | 0.7137 | 0.7145 | 0.7155 | 0.7135 | 0.7117 | 0.7150 | 0.7127 | 0.7111 | 0.7147 | 0.7136 |
| | AdagradAR | **0.7253** | **0.7247** | **0.7234** | **0.7241** | **0.7269** | **0.7258** | **0.7251** | **0.7257** | **0.7273** | **0.7268** | **0.7264** | **0.7263** | **0.7269** | **0.7260** | **0.7252** | **0.7239** |

Figure 2(a) demonstrates that as $r$ decreases, the test AUC remains stable across multiple epochs, indicating that the one-epoch overfitting phenomenon is effectively alleviated. It suggests that low-frequency IDs in the "IP" feature are the primary cause of one-epoch overfitting in the iPinYou dataset. However, as shown by the case $r = 0$, removing the "IP" feature results in a substantial test AUC drop (from 0.7498 to 0.7429) at the end of the first epoch.

Although removing sparse features can mitigate the one-epoch overfitting issue, it often leads to diminished model performance. In contrast, our proposed method dynamically adjusts the regularization strength, effectively preventing overfitting while maintaining strong predictive performance.

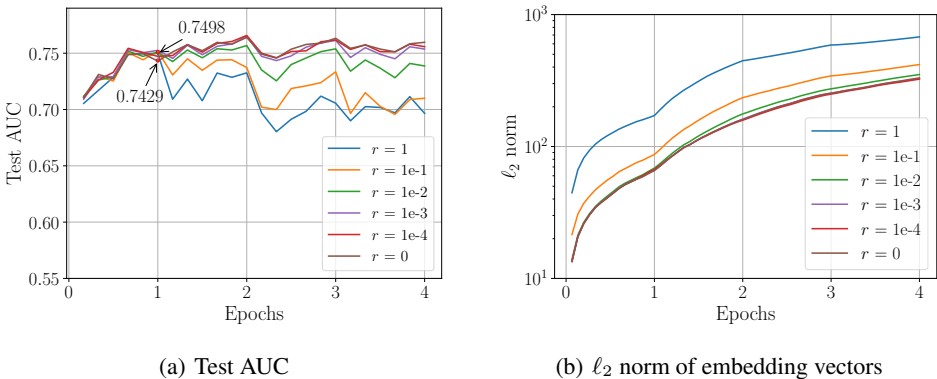

(a) Test AUC  (b) $\ell_2$ norm of embedding vectors

Figure 2: Performance comparison using various filter ratios for the "IP" feature on the iPinYou dataset. (a) shows the test AUC results. (b) presents the cumulative $\ell_2$ norm of embedding vectors.

## 4.5 ABLATION STUDY AND BUCKET ANALYSIS

We use the iPinYou dataset for analysis because it contains a single feature causing the one-epoch issue as described in section 4.4. Since "IP" can be interpreted as a proxy for a user, we create 5

buckets based on the frequency of the "IP" feature to perform user-based bucket analysis, examining AUC gains, regularization strength, and $\ell_2$ norm of our method. A smaller bucket index indicates a lower occurrence frequency. Figure 3 shows that the bucket norms can be controlled via adaptive regularization strength while preserving the AUC gains across all buckets, and our method achieves particularly strong performance in the high-frequency bucket due to the larger norm budget available. Then, we conduct an ablation study on occurrence interval estimation with a DNN backbone. In the AdamW baseline, we apply a constant weight decay to the embedding layers only. We then gradually apply our method to different buckets from 1 to 5. Table 3 shows that, compared with using a constant weight decay, decreasing the weight decay for high-frequency features and increasing it for low-frequency features can further improve performance while alleviating the one-epoch issue.

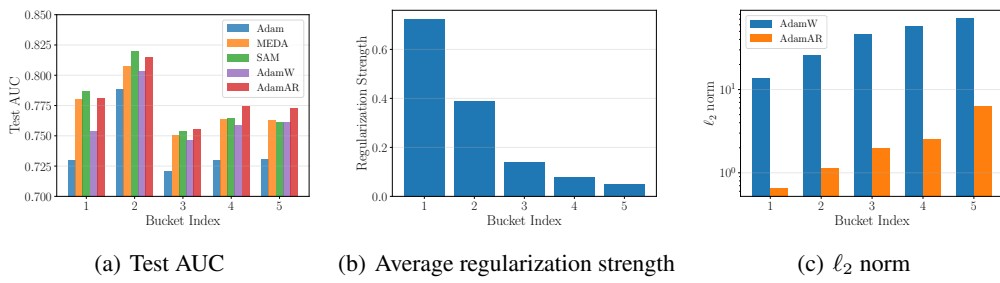

(a) Test AUC       (b) Average regularization strength      (c) $\ell_2$ norm

Figure 3: Bucket analysis of "IP" feature on the iPinYou dataset with DNN backbone and Adam optimizer. (a) shows the test AUC results over different feature frequency buckets at the end of epoch 2. (b) presents the regularization strength. (c) shows the cumulative $\ell_2$ norm of "IP" feature at the end of epoch 2.

Table 3: Comparison of test AUC for ablation study using Adam optimizer.

| Experiment Setting | E1 | E2 | E3 | E4 |
|---|---|---|---|---|
| AdamW | 0.7486 | 0.7595 | 0.7628 | 0.7500 |
| AdamAR-Bucket 1 & AdamW-Bucket 2-5 | 0.7457 | 0.7607 | 0.7646 | 0.7520 |
| AdamAR-Bucket 1-2 & AdamW-Bucket 3-5 | 0.7496 | 0.7622 | 0.7655 | 0.7556 |
| AdamAR-Bucket 1-3 & AdamW-Bucket 4-5 | 0.7510 | 0.7646 | 0.7657 | 0.7614 |
| AdamAR-Bucket 1-4 & AdamW-Bucket 5 | 0.7470 | 0.7656 | 0.7660 | 0.7635 |
| AdamAR | 0.7549 | 0.7728 | 0.7730 | 0.7725 |

## 5 CONCLUSION

We propose an adaptive regularization method to address the one-epoch problem in estimation models for the ASR domain. Experimental results demonstrate that our approach effectively mitigates the one-epoch issue and improves estimation performance. Additionally, we provide a theoretical explanation for the one-epoch phenomenon and illustrate how the proposed method takes effect through analytical derivations and experiments. This approach has already been fully deployed in the production environment of sponsored search in our company.

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

APPENDIX

# A GenAI Usage Disclosure

During the preparation of this work, the author used ChatGPT to improve the language. After using this tool, the author reviewed and edited the content as necessary and takes full responsibility for the final publication.

# B Minimum Convergence Analysis

## B.1 Proof of Minimum Convergence

Based on the assumptions outlined in section 3.3, and following the analytical framework of He et al. (2023), we apply the descent lemma (Nesterov (2013)) to derive

$$
\begin{aligned}
f\left(\boldsymbol{\theta}^{k+1}\right) &\leq f\left(\boldsymbol{\theta}^k\right) + \left\langle \nabla f(\boldsymbol{\theta}^k), \boldsymbol{\theta}^{k+1} - \boldsymbol{\theta}^k \right\rangle + \frac{L_0}{2} \left\| \boldsymbol{\theta}^{k+1} - \boldsymbol{\theta}^k \right\|^2 \\
&\overset{\text{(a)}}{=} f(\boldsymbol{\theta}^k) - \eta_k \left\langle \nabla f(\boldsymbol{\theta}^k), \frac{\boldsymbol{m}^k}{\sqrt{\boldsymbol{v}^k + \varepsilon}} + \boldsymbol{\lambda}_0^k \boldsymbol{\theta}^k \right\rangle + \frac{L_0 \eta_k^2}{2} \left\| \frac{\boldsymbol{m}^k}{\sqrt{\boldsymbol{v}^k + \varepsilon}} + \boldsymbol{\lambda}_0^k \boldsymbol{\theta}^k \right\|^2 \\
&\leq f(\boldsymbol{\theta}^k) - \eta_k \left\langle \nabla f(\boldsymbol{\theta}^k), \frac{\boldsymbol{m}^k}{\sqrt{\boldsymbol{v}^k + \varepsilon}} \right\rangle + \frac{L_0 \eta_k^2}{2} \left\| \frac{\boldsymbol{m}^k}{\sqrt{\boldsymbol{v}^k + \varepsilon}} \right\|^2 \\
&\quad - \eta_k \left\langle \nabla f(\boldsymbol{\theta}^k), \boldsymbol{\lambda}_0^k \boldsymbol{\theta}^k \right\rangle + \frac{L_0 \eta_k^2}{2} \left\| \frac{\boldsymbol{m}^k}{\sqrt{\boldsymbol{v}^k + \varepsilon}} \right\|^2 + L_0 \eta_k^2 \left\| \boldsymbol{\lambda}_0^k \right\|^2 \left\| \boldsymbol{\theta}^k \right\|^2
\end{aligned}
\tag{16}
$$

where $\eta_k = \eta \frac{\sqrt{(1-\beta_2^k)}}{1-\beta_1^k}$ and $\eta$ can be specified either as a constant or according to a schedule. Let $\boldsymbol{\lambda}_0^k = \min(1, \alpha \boldsymbol{I}^k)/\eta_k$, we can then construct $\boldsymbol{\lambda}^k = \eta_k \boldsymbol{\lambda}_0^k$ so that $(a)$ is satisfied. Based on proposition 2 and Lemma 16 in He et al. (2023), we can derive $\mathbb{E}\left[\|\boldsymbol{\theta}^k\|\right] \leq W, \forall k \geq 1$ by mathematical induction, where $W \in \mathbb{R}$ is a constant. Furthermore, according to appendix B.2, $\mathbb{E}\left[\|\bar{\boldsymbol{I}} + \boldsymbol{\delta}'^k\|\right] \leq \|\bar{\boldsymbol{I}}\| + \nu_2$ and $\eta_k$ is lower bounded by $\eta_{min}$, we can derive that

$$
\mathbb{E}\left[\|\boldsymbol{\lambda}_0^k\|\right] \leq \alpha \left(\|\bar{\boldsymbol{I}}\| + \nu_2\right)/\eta_{min}
\tag{17}
$$

Let $\alpha \left(\|\bar{\boldsymbol{I}}\| + \nu_2\right)/\eta_{min} = \nu_0$, and applying the Cauchy-Schwarz inequality yields

$$
\mathbb{E}\left[\langle \nabla f(\boldsymbol{\theta}^k), \boldsymbol{\lambda}_0^k \boldsymbol{\theta}^k \rangle\right] \geq -\mathbb{E}\left[\|\nabla f(\boldsymbol{\theta}^k)\| \|\boldsymbol{\lambda}_0^k\| \|\boldsymbol{\theta}^k\|\right] \geq -M\nu_0 W
\tag{18}
$$

Then we invoke the equation 23 in He et al. (2023), under the conditions $0 < \beta_1 < 1$ and $0 < \beta_2 < 1$

$$
\begin{aligned}
\mathbb{E}\left[f\left(\boldsymbol{\theta}^{k+1}\right)\right] &\leq \mathbb{E}\left[f\left(\boldsymbol{\theta}^k\right)\right] - \frac{1-\beta_1}{\sqrt{M^2 + \varepsilon}} \eta_k \mathbb{E}\left[\left\|\nabla f\left(\boldsymbol{\theta}^k\right)\right\|^2\right] \\
&\quad + \frac{\beta_1 L_0 M^2}{\varepsilon} \eta_k \sum_{i=1}^k \beta_1^{k-i} \eta_{i-1} + \frac{\sqrt{P} M^4}{\varepsilon^{3/2}} \eta_k \sum_{i=1}^k \beta_1^{k-i}(1-\beta_2) \\
&\quad + \frac{M^2 L_0}{\varepsilon} \eta_k^2 + M\nu_0 W \eta_k + L_0 \nu_0^2 W^2 \eta_k^2.
\end{aligned}
\tag{19}
$$

Upon rearranging the above equation and summing both sides over $k$ from 1 to $T$, we have

$$
\begin{aligned}
\frac{1-\beta_1}{\sqrt{M^2+\varepsilon}} \sum_{k=1}^K \eta_k \mathbb{E}\left[\left\|\nabla f\left(\boldsymbol{\theta}^k\right)\right\|^2\right] &\leq f(\boldsymbol{\theta}^1) - f^* + \frac{\beta_1 L_0 M^2}{\varepsilon} \sum_{k=1}^K \eta_k \sum_{i=1}^k \beta_1^{k-i} \eta_{i-1} \\
+ \frac{\sqrt{P} M^4}{\varepsilon^{3/2}} \sum_{k=1}^K \eta_k \sum_{i=1}^k \beta_1^{k-i}(1-\beta_2) &+ \frac{M^2 L_0}{\varepsilon} \sum_{k=1}^K \eta_k^2 + M\nu_0 W \sum_{k=1}^K \eta_k + L_0 \nu_0^2 W^2 \sum_{k=1}^K \eta_k^2.
\end{aligned}
\tag{20}
$$

Suppose that $\{\gamma^k\}_{k \geq 1}$ is a non-increasing real sequence. Assume that there exist positive constants $C_0$ and $\tilde{C}_0$ such that $C_0 \gamma_k \leq \eta_k \leq \tilde{C}_0 \gamma_k$. By applying the equation 25 and 26 in He et al. (2023), we obtain

$$\frac{1 - \beta_1}{\sqrt{M^2 + \varepsilon}} \sum_{k=1}^{K} \eta_k \mathbb{E}\left[\left\|\nabla f\left(\boldsymbol{\theta}^k\right)\right\|^2\right] \leq f\left(\boldsymbol{\theta}^1\right) - f^* + \frac{\tilde{C}_0^2 \beta_1 L_0 M^2}{\varepsilon\, C_0^2(1 - \beta_1)} \sum_{k=1}^{K} \eta_k^2$$

$$+ \frac{\tilde{C}_0 \sqrt{P}\, M^4 (1 - \beta_2)}{\varepsilon^{3/2} C_0 (1 - \beta_1)} \sum_{k=1}^{K} \eta_k + \frac{M^2 L_0}{\varepsilon} \sum_{k=1}^{K} \eta_k^2 + M \nu_0 W \sum_{k=1}^{K} \eta_k + L_0 \nu_0^2 W^2 \sum_{k=1}^{K} \eta_k^2. \tag{21}$$

multiplying both sides of the above equation by $\frac{\sqrt{M^2 + \varepsilon}}{1 - \beta_1}$, we finally obtain

$$\sum_{k=1}^{K} \eta_k \mathbb{E}\left[\left\|\nabla f\left(\boldsymbol{\theta}^k\right)\right\|^2\right] \leq C_1 + C_2 \sum_{k=1}^{K} \eta_k + C_3 \sum_{k=1}^{K} \eta_k^2 \tag{22}$$

where $C_1$, $C_2$ and $C_3$ in equation 22 are given by

$$C_1 = \frac{\sqrt{M^2 + \varepsilon}\, (f(\boldsymbol{\theta}^1) - f^*)}{1 - \beta_1},$$

$$C_2 = \frac{\tilde{C}_0 M^4 (1 - \beta_2) \sqrt{P(M^2 + \varepsilon)}}{\varepsilon^{3/2} C_0 (1 - \beta_1)^2} + \frac{M \nu_0 W \sqrt{(M^2 + \varepsilon)}}{1 - \beta_1}, \tag{23}$$

$$C_3 = \frac{\tilde{C}_0^2 \beta_1 M^2 L_0 \sqrt{M^2 + \varepsilon}}{\varepsilon\, C_0^2 (1 - \beta_1)^2} + \frac{M^2 L_0 \sqrt{M^2 + \varepsilon}}{\varepsilon (1 - \beta_1)} + \frac{L_0 \nu_0^2 W^2 \sqrt{M^2 + \varepsilon}}{1 - \beta_1}.$$

because of

$$\min_{1 \leq k \leq K} \mathbb{E}\left[\left\|\nabla f\left(\boldsymbol{\theta}^k\right)\right\|^2\right] \sum_{k=1}^{K} \eta_k \leq \sum_{k=1}^{K} \eta_k \mathbb{E}\left[\left\|\nabla f\left(\boldsymbol{\theta}^k\right)\right\|^2\right] \tag{24}$$

we can derive that

$$\min_{1 \leq k \leq K} \mathbb{E}\left[\left\|\nabla f\left(\boldsymbol{\theta}^k\right)\right\|^2\right] \leq \frac{C_1 + C_2 \sum_{k=1}^{K} \eta_k + C_3 \sum_{k=1}^{K} \eta_k^2}{\sum_{k=1}^{K} \eta_k} \tag{25}$$

It can be observed that the incorporation of adaptive regularization modifies only the constant term in the upper bound of the minimum convergence rate, while leaving the inherent convergence properties of the Adam algorithm unaffected.

## B.2 BOUND OF OCCURRENCE INTERVAL NOISE

We assume the occurrence interval $\boldsymbol{I}^k$ is subject to additive random noise $\boldsymbol{\delta}^k$, and it holds that $\mathbb{E}[\boldsymbol{I}^k] = \bar{\boldsymbol{I}} + \boldsymbol{\nu}_1$, where $\mathbb{E}\left[\boldsymbol{\delta}^k\right] = \boldsymbol{\nu}_1$, $\bar{\boldsymbol{I}} \in \mathbb{R}^P$ is finite constants vector. Let $\boldsymbol{I}^k = \bar{\boldsymbol{I}} + \boldsymbol{\delta}^k$. Since $\boldsymbol{I}^k \geq 0$, the random noise $\boldsymbol{\delta}^k$ has a lower bound of $-\bar{\boldsymbol{I}}$. We rewrite the vector form of equation 11 as

$$\boldsymbol{\lambda}^k = \min\left(1, \alpha\left(\bar{\boldsymbol{I}} + \boldsymbol{\delta}^k\right)\right) = \alpha(\bar{\boldsymbol{I}} + \min\left(\boldsymbol{\delta}^k, 1/\alpha - \bar{\boldsymbol{I}}\right)) \tag{26}$$

Let $\boldsymbol{\delta}'^k = \min\left(\boldsymbol{\delta}^k, 1/\alpha - \bar{\boldsymbol{I}}\right)$, then it is straightforward to show that the actual occurrence interval noise is bounded, i.e., $\boldsymbol{\delta}'^k \in [-\bar{\boldsymbol{I}}, 1/\alpha - \bar{\boldsymbol{I}}]$. So we can see that the actual noise which impact $\boldsymbol{\lambda}^k$ has been bounded, i.e., $\mathbb{E}\left[\|\boldsymbol{\delta}'^k\|^2\right] \leq \nu_2^2$ and $\nu_2$ is finite constant.

## C NON-SMOOTH CONDITION DISCUSSION

We analyze the non-smooth condition under the assumption that the function $\varphi(\tau_{ij})$ is locally Lipschitz continuous. Based on the Clarke subdifferential (Clarke (1975)) and the generalized stationarity condition for the optimal solution of the inner problem, we obtain

$$-\lambda_{ij}^* \in \partial\phi(\tau^*) \tag{27}$$

where $\partial\phi(\tau^*)$ denotes the set of subgradients at $\tau^*$. Based on generalized KKT framework, there exists $\rho_{ij} \in \partial\phi(\tau^*)$ such that

$$m_{ij}\rho_{ij} + \mu_0 = 0 \tag{28}$$

Assuming that $\partial\phi(\tau^*) \cap \{\rho \,|\, m_{ij}\rho + \mu_0 = 0\} \neq \varnothing$, we can select a consistent subgradient from $\partial\phi(\tau^*)$ such that $\rho_{ij} = -\lambda_{ij}^*$. Substituting this choice into equation 28 yields $\lambda_{ij}^* = \mu_0/m_{ij}$. Therefore, the necessary condition remains valid in the non-smooth case.

## D    RADEMACHER COMPLEXITY BOUND OF FM MODEL

We analyze the Rademacher complexity of a bias-free factorization machine (FM) model. Let $f_{FM} \in \mathcal{F}$ be defined as

$$f_{FM}([e_1(t); e_2(t); \dots; e_S(t)]) = \sum_{i=1}^{S} w_i^\top e_i(t) + \sum_{i=1}^{S} \sum_{j=i+1}^{S} e_i(t)^\top e_j(t) \tag{29}$$

where $\mathcal{F}$ denotes the class of real-valued FM like functions, Let $w_i \in \mathbb{R}^{d_i}, \forall i \in S$ denote the linear weight vector and let $e_i(t)$ be the embedding vector for feature $i$ at $t$. We decompose the Rademacher complexity into linear part $\widehat{\mathcal{R}}_T(\mathcal{F}_l)$ and interaction part $\widehat{\mathcal{R}}_T(\mathcal{F}_q)$, where $\mathcal{F}_l$ define the linear function class and $\mathcal{F}_q$ define the interaction function class. Therefore, the overall upper bound of $\widehat{\mathcal{R}}_T(\mathcal{F})$ satisfy

$$\widehat{\mathcal{R}}_T(\mathcal{F}) \leq \widehat{\mathcal{R}}_T(\mathcal{F}_l) + \widehat{\mathcal{R}}_T(\mathcal{F}_q) \tag{30}$$

For the linear part, the upper bound of $\widehat{\mathcal{R}}_T(\mathcal{F}_l)$ can be obtained as

$$\widehat{\mathcal{R}}_T(\mathcal{F}_l) \leq \frac{1}{T} \sum_{i=1}^{S} \|w_i\| \sqrt{\sum_{t=1}^{T} \|e_i(t)\|^2} \leq \frac{1}{\sqrt{T}} \sum_{i=1}^{S} \|w_i\| M_{E_i} \tag{31}$$

According to Bartlett & Mendelson (2002), the upper bound of $\widehat{\mathcal{R}}_T(\mathcal{F}_q)$ is given by

$$\widehat{\mathcal{R}}_T(\mathcal{F}_q) \leq \frac{1}{\sqrt{T}} \sum_{i=1}^{S} \sum_{j=i+1}^{S} M_{E_i} M_{E_j} \tag{32}$$

Finally, we obtain the upper bound of the Rademacher complexity for the FM-like function class as follows

$$\widehat{\mathcal{R}}_T(\mathcal{F}) \leq \frac{1}{\sqrt{T}} \left( \sum_{i=1}^{S} \|w_i\| M_{E_i} + \sum_{i=1}^{S} \sum_{j=i+1}^{S} M_{E_i} M_{E_j} \right) \tag{33}$$

From equation 33, we observe that the embedding norm plays the most important role in determining the upper bound of the Rademacher complexity and ultimately affects the generalization error.

## E    ANALYSIS OF EMBEDDING SIZE

We examine the impact of varying embedding sizes on the xDeepFM backbone using the iPinYou dataset. The embedding size is ranged from 16 to 128, and the multi-epoch performance is compared across five methods. As shown in table 4, our method performs consistently well across all embedding sizes. Furthermore, we examine the singular spectrum of the embedding matrix for the "IP" feature. As shown in figure 4(a), training with the basic Adam optimizer leads to a rapid increase in the sum of singular values (SS) across multiple epochs. In contrast, figure 4(b) demonstrates that the information abundance (IA), as defined in Guo et al. (2023), decreases correspondingly. This inverse relationship reveals the occurrence of an embedding-collapse phenomenon. Notably, our method effectively controls SS growth and enhances IA, which may explain its superior performance in mitigating overfitting.

## F    COMPUTAIONT AND MEMORY ANALYSIS

We compare the computation and memory costs of AdamAR and Adam, excluding gradient computation. $P$ is number of parameters. The total costs are summarized in the table below.

Table 4: Comparison of test AUC with xDeepFM backbone and iPinYou dataset using Adam optimizer across different embedding sizes. The best results are highlighted in bold.

| Method | Embedding Size = 16 | | | | Embedding Size = 32 | | | | Embedding Size = 64 | | | | Embedding Size = 128 | | | |
|---|---|---|---|---|---|---|---|---|---|---|---|---|---|---|---|---|
| | E1 | E2 | E3 | E4 | E1 | E2 | E3 | E4 | E1 | E2 | E3 | E4 | E1 | E2 | E3 | E4 |
| Adam | 0.7593 | 0.7373 | 0.6914 | 0.6980 | 0.7607 | 0.7379 | 0.6928 | 0.6831 | 0.7579 | 0.7396 | 0.7158 | 0.6867 | 0.7630 | 0.7430 | 0.7076 | 0.6815 |
| MEDA | 0.7593 | 0.7603 | 0.7548 | 0.7628 | 0.7607 | 0.7592 | 0.7544 | 0.7595 | 0.7579 | 0.7596 | 0.7624 | 0.7594 | 0.7630 | 0.7580 | 0.7603 | 0.7627 |
| SAM | 0.7602 | 0.7567 | 0.7408 | 0.7396 | 0.7644 | 0.7514 | 0.7401 | 0.7265 | 0.7582 | 0.7516 | 0.7385 | 0.7197 | 0.7680 | 0.7526 | 0.7394 | 0.7200 |
| AdamW | 0.7622 | 0.7683 | 0.7653 | 0.7632 | 0.7611 | 0.7655 | 0.7652 | 0.7588 | 0.7615 | 0.7663 | 0.7638 | 0.7274 | 0.7701 | 0.7691 | 0.7691 | 0.7475 |
| AdamAR | **0.7688** | **0.7757** | **0.7710** | **0.7700** | **0.7744** | **0.7732** | **0.7708** | **0.7681** | **0.7668** | **0.7733** | **0.7698** | **0.7653** | **0.7770** | **0.7719** | **0.7721** | **0.7646** |

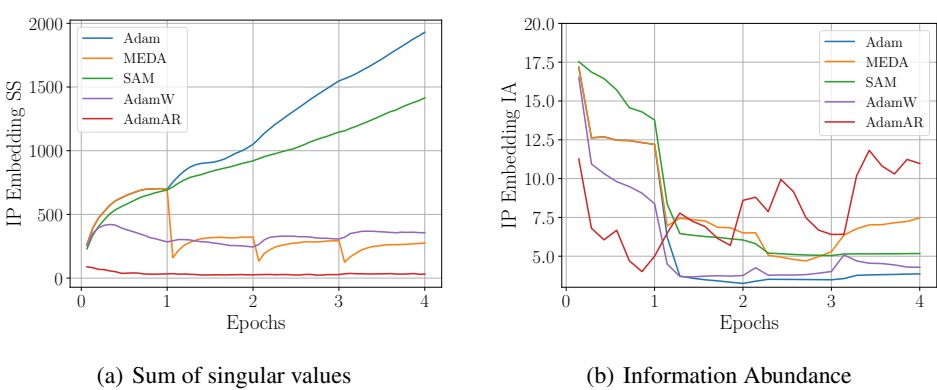

(a) Sum of singular values

(b) Information Abundance

Figure 4: Singular spectrum analysis of "IP" feature embedding on the iPinYou dataset. (a) shows the sum of singular values. (b) presents the information abundance.

## G    ADAGRAD WITH ADAPTIVE REGULARIZATION

Adagrad, like Adam, is widely used in the ASR domain. Our method extends readily to Adagrad, as detailed in Algorithm 2.

---

**Algorithm 2** Adagrad with Adaptive Regularization (AdagradAR)

---

1: given $\varepsilon = 10^{-8}$, $\alpha$, learning rate $\eta$
2: initialize time step $t \leftarrow 0$, parameter $\theta_p^{t=0}$, squared gradient accumulator $v_p^{t=0} \leftarrow 0$, last update step state $s_p^{t=0} \leftarrow 0$
3: **repeat**
4:     $t \leftarrow t + 1$
5:     $g_p^t \leftarrow \nabla_{\theta_p} f\left(\theta_p^{t-1}\right)$
6:     $v_p^t \leftarrow v_p^{t-1} + \left(g_p^t\right)^2$
7:     $\lambda_p^t \leftarrow \min\left(1, \left(t - s_p^{t-1} - 1\right)\alpha\right)$
8:     $s_p^t \leftarrow t$ if $||g_p^t|| > 0$ else $s_p^{t-1}$
9:     $\theta_p^t \leftarrow \theta_p^{t-1} - \lambda_p^t \theta_p^{t-1} - \eta \cdot g_p^t / \left(\sqrt{v_p^t} + \varepsilon\right)$
10: **until** stopping criterion is met
11: return optimized parameters $\theta_p^t$

---

## H    NETWORK ARCHITECTURE CONFIGURATIONS

We adjust the network architectures based on the feature and sample sizes of each dataset. As summarized in table 6, larger and deeper networks are utilized for datasets with more features and samples, such as iPinYou, Avazu, and LZD. In contrast, smaller and shallower networks are adopted for the Amazon dataset to mitigate the risk of overfitting.

Table 5: Comparison of per-iteration computation and memory costs between Adam and AdamAR.

| Optimizer | Mul. / iter | Add. / iter | Memory cost |
|---|---|---|---|
| Adam | $\approx 9P$ | $\approx 3P$ | $3P$ |
| AdamAR | $\approx 11P$ | $\approx 5P$ | $4P$ |

Table 6: Network architecture configurations for different datasets

| Dataset | DNN | WDL | xDeepFM | | | WuKong | | |
|---|---|---|---|---|---|---|---|---|
| | MLP Hiddens | MLP Hiddens | CIN Hiddens | MLP Hiddens | Layers | LCB&FMB Embs | FMB Hiddens | MLP Hiddens |
| iPinYou | [512, 256, 128] | [512, 256, 128] | [16, 16, 16] | [512, 256, 128] | 5 | 12 | [64, 32] | [512, 256, 128] |
| Amazon | [512, 256, 128] | [512, 256] | [8, 8] | [512, 256] | 1 | 8 | [32, 32] | [512, 256] |
| Avazu | [512, 256, 128] | [512, 256, 128] | [16, 16, 16] | [512, 256, 128] | 5 | 12 | [64, 32] | [512, 256, 128] |
| LZD | [512, 256, 128] | [512, 256, 128] | [16, 16, 16] | [512, 256, 128] | 5 | 12 | [64, 32] | [512, 256, 128] |

# I GRID SEARCH FOR THE WEIGHT DECAY COEFFICIENT

The weight decay coefficient is a crucial hyperparameter that can significantly impact model performance. In our adaptive regularization method, this coefficient is denoted by $\alpha$ in equation 11. To ensure fair comparisons across different methods, we perform a grid search to identify the optimal weight decay value for each dataset. Figures 5, 6, 7, and 8 illustrate the effect of varying the weight decay coefficient on test AUC across all datasets using a DNN backbone at the end of epoch 2. Although test AUC varies across weight decay coefficient settings, our proposed method consistently surpasses the performance of AdamW and AdagradW. Moreover, whereas AdamW and AdagradW exhibit high sensitivity to the weight decay coefficient, our method maintains stable performance, thereby reducing the complexity of hyperparameter tuning in practice. Notably, the Avazu and LZD datasets, which contain a larger number of sparse features and samples, require a smaller optimal weight decay coefficient than the iPinYou and Amazon datasets. This observation offers practical guidance for coefficient selection in real-world industrial scenarios.

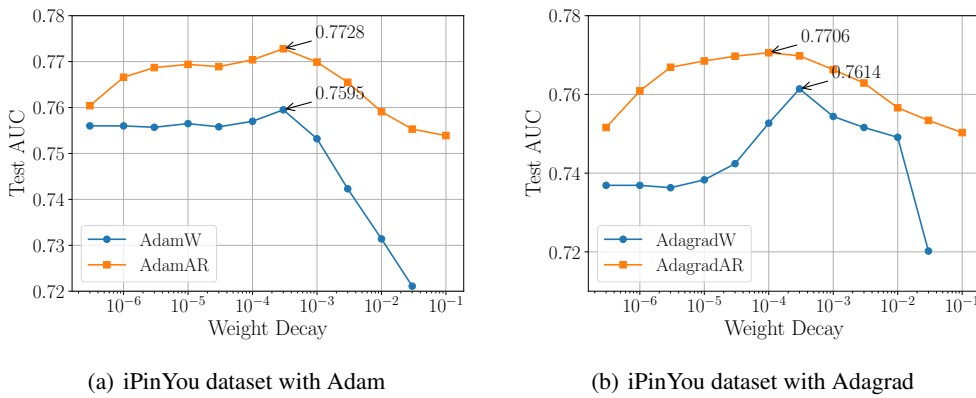

(a) iPinYou dataset with Adam          (b) iPinYou dataset with Adagrad

Figure 5: Performance of different weight decay coefficient on iPinYou dataset with DNN backbone at the end of epoch 2. (a) shows the performance with Adam. (b) shows the performance with Adagrad.

# J DETAILED EXPERIMENTAL RESULTS

Here we present detailed experimental results with estimated standard deviation in section 4.3.

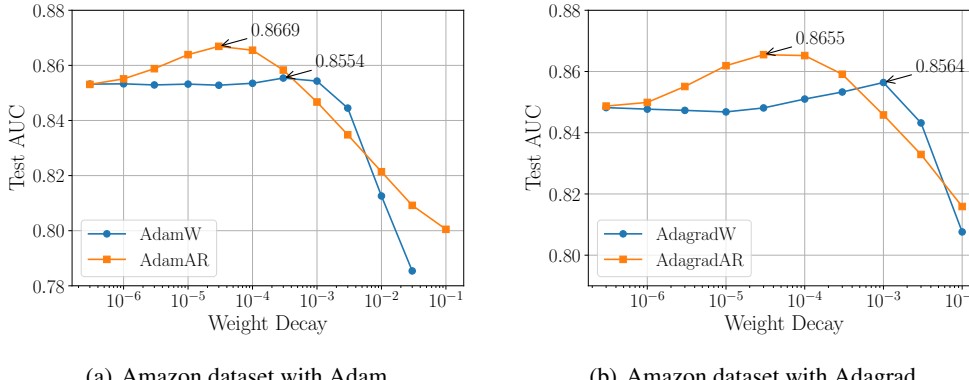

(a) Amazon dataset with Adam        (b) Amazon dataset with Adagrad

Figure 6: Performance of different weight decay coefficient on Amazon dataset with DNN backbone at the end of epoch 2. (a) shows the performance with Adam. (b) shows the performance with Adagrad.

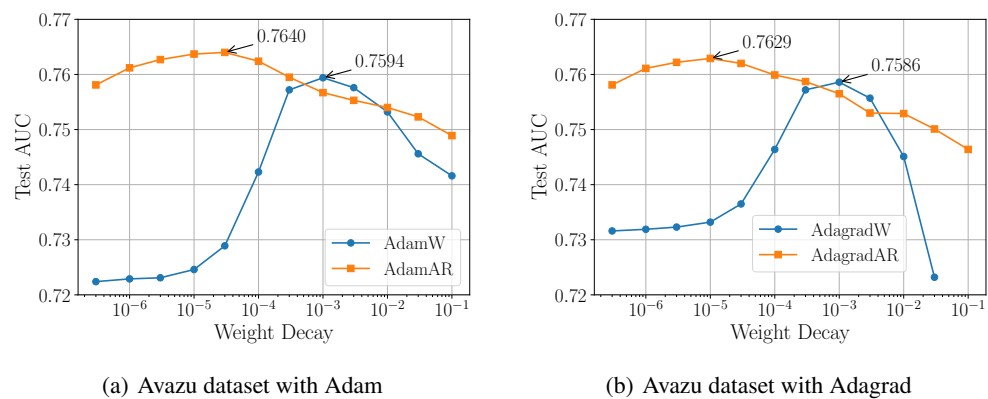

(a) Avazu dataset with Adam        (b) Avazu dataset with Adagrad

Figure 7: Performance of different weight decay coefficient on Avazu dataset with DNN backbone at the end of epoch 2. (a) shows the performance with Adam. (b) shows the performance with Adagrad.

## K  FEATURE STATISTICS ON IPINYOU DATASET

The iPinYou dataset contains a total of 16 features, among which only a few are sparse. Table 7 lists the top six sparse features along with their statistical indicators.

## L  DATASET DETAILS

**iPinYou**. The training dataset includes processed bidding, impression, click, and conversion logs from iPinYou DSP. It contains 19.5 million records and 16 categorical features, providing a suitable context for exemplifying the overfitting phenomenon.

**Amazon**. This is a widely used dataset from Amazon for evaluating CTR estimation models. In our study, we use the electronics category of the Amazon dataset, which contains approximately 3 million records and 3 categorical features.

**Avazu**. This dataset comprises approximately 10 days of labeled click-through data from mobile advertisements, consisting of 40 million records and 22 categorical features spanning both user attributes and advertisement characteristics.

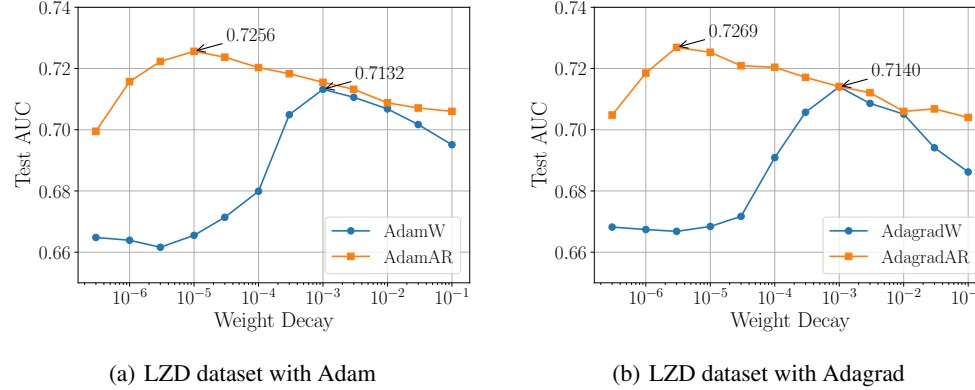

(a) LZD dataset with Adam     (b) LZD dataset with Adagrad

Figure 8: Performance of different weight decay coefficient on LZD dataset with DNN backbone at the end of epoch 2. (a) shows the performance with Adam. (b) shows the performance with Adagrad.

Table 7: For the top six sparse features in the iPinYou dataset, the table below presents the number of unique IDs, the average occurrences of each ID, and the mean update interval for each ID with a batch size of 2048.

| Feature | IP | slotid | domain | city | creative | useragent |
|---|---|---|---|---|---|---|
| Unique IDs | 704,966 | 180,696 | 51,322 | 372 | 133 | 42 |
| Mean Occurrences | 27.7 | 107.9 | 379.9 | 52,408.5 | 146,586.3 | 464,189.9 |
| Mean Update Intervals | 344.222 | 88.230 | 25.060 | 0.182 | 0.065 | 0.021 |

**LZD**. The dataset is sampled from our production environment and consists of real-time bidding logs from sponsored search. The dataset contains 25 million records and 13 categorical features.

Table 8: Comparison of average test AUC with DNN and WDL models using Adam optimizer.

| Dataset | Method | DNN | | | | WDL | | | |
|---|---|---|---|---|---|---|---|---|---|
| | | E1 | E2 | E3 | E4 | E1 | E2 | E3 | E4 |
| iPinYou | Adam | $0.7515_{\pm0.0023}$ | $0.7304_{\pm0.0050}$ | $0.7061_{\pm0.0109}$ | $0.7014_{\pm0.0026}$ | $0.7619_{\pm0.0006}$ | $0.7320_{\pm0.0028}$ | $0.7028_{\pm0.0101}$ | $0.6987_{\pm0.0041}$ |
| | MEDA | $0.7515_{\pm0.0023}$ | $0.7644_{\pm0.0051}$ | $0.7684_{\pm0.0043}$ | $0.7717_{\pm0.0022}$ | $0.7619_{\pm0.0006}$ | $0.7589_{\pm0.0013}$ | $0.7565_{\pm0.0017}$ | $0.7551_{\pm0.0045}$ |
| | SAM | $0.7510_{\pm0.0078}$ | $0.7593_{\pm0.0072}$ | $0.7445_{\pm0.0099}$ | $0.7256_{\pm0.0058}$ | $0.7610_{\pm0.0047}$ | $0.7581_{\pm0.0010}$ | $0.7404_{\pm0.0019}$ | $0.7248_{\pm0.0036}$ |
| | AdamW | $0.7475_{\pm0.0014}$ | $0.7592_{\pm0.0087}$ | $0.7623_{\pm0.0030}$ | $0.7568_{\pm0.0061}$ | $0.7551_{\pm0.0111}$ | $0.7656_{\pm0.0049}$ | $0.7646_{\pm0.0056}$ | $0.7634_{\pm0.0019}$ |
| | AdamAR | $\mathbf{0.7566_{\pm0.0019}}$ | $\mathbf{0.7692_{\pm0.0076}}$ | $\mathbf{0.7688_{\pm0.0079}}$ | $\mathbf{0.7724_{\pm0.0002}}$ | $\mathbf{0.7655_{\pm0.0063}}$ | $\mathbf{0.7729_{\pm0.0026}}$ | $\mathbf{0.7670_{\pm0.0040}}$ | $\mathbf{0.7668_{\pm0.0016}}$ |
| Amazon | Adam | $0.8482_{\pm0.0007}$ | $0.8548_{\pm0.0014}$ | $0.8335_{\pm0.0019}$ | $0.8180_{\pm0.0011}$ | $0.8474_{\pm0.0016}$ | $0.8510_{\pm0.0017}$ | $0.8261_{\pm0.0009}$ | $0.8156_{\pm0.0029}$ |
| | MEDA | $0.8482_{\pm0.0007}$ | $0.8544_{\pm0.0011}$ | $0.8556_{\pm0.0008}$ | $0.8573_{\pm0.0003}$ | $0.8474_{\pm0.0016}$ | $0.8506_{\pm0.0017}$ | $0.8566_{\pm0.0002}$ | $0.8566_{\pm0.0007}$ |
| | SAM | $0.8507_{\pm0.0021}$ | $0.8587_{\pm0.0025}$ | $0.8417_{\pm0.0018}$ | $0.8249_{\pm0.0012}$ | $\mathbf{0.8516_{\pm0.0016}}$ | $0.8567_{\pm0.0006}$ | $0.8396_{\pm0.0010}$ | $0.8213_{\pm0.0023}$ |
| | AdamW | $0.8476_{\pm0.0008}$ | $0.8571_{\pm0.0017}$ | $0.8426_{\pm0.0018}$ | $0.8276_{\pm0.0010}$ | $0.8461_{\pm0.0013}$ | $0.8533_{\pm0.0015}$ | $0.8380_{\pm0.0003}$ | $0.8223_{\pm0.0024}$ |
| | AdamAR | $\mathbf{0.8507_{\pm0.0010}}$ | $\mathbf{0.8683_{\pm0.0015}}$ | $\mathbf{0.8708_{\pm0.0005}}$ | $\mathbf{0.8686_{\pm0.0008}}$ | $0.8496_{\pm0.0017}$ | $\mathbf{0.8654_{\pm0.0011}}$ | $\mathbf{0.8689_{\pm0.0002}}$ | $\mathbf{0.8659_{\pm0.0014}}$ |
| Avazu | Adam | $0.7461_{\pm0.0031}$ | $0.7205_{\pm0.0022}$ | $0.7014_{\pm0.0035}$ | $0.6883_{\pm0.0018}$ | $0.7483_{\pm0.0013}$ | $0.7221_{\pm0.0012}$ | $0.6982_{\pm0.0029}$ | $0.6886_{\pm0.0022}$ |
| | MEDA | $0.7461_{\pm0.0031}$ | $0.7498_{\pm0.0010}$ | $0.7489_{\pm0.0009}$ | $0.7485_{\pm0.0012}$ | $0.7483_{\pm0.0013}$ | $0.7488_{\pm0.0012}$ | $0.7480_{\pm0.0008}$ | $0.7494_{\pm0.0021}$ |
| | SAM | $0.7451_{\pm0.0019}$ | $0.7194_{\pm0.0028}$ | $0.7013_{\pm0.0031}$ | $0.6899_{\pm0.0030}$ | $0.7477_{\pm0.0013}$ | $0.7205_{\pm0.0026}$ | $0.6993_{\pm0.0004}$ | $0.6902_{\pm0.0005}$ |
| | AdamW | $0.7572_{\pm0.0011}$ | $0.7582_{\pm0.0011}$ | $0.7582_{\pm0.0006}$ | $0.7583_{\pm0.0007}$ | $0.7585_{\pm0.0008}$ | $0.7581_{\pm0.0014}$ | $0.7563_{\pm0.0017}$ | $0.7570_{\pm0.0011}$ |
| | AdamAR | $\mathbf{0.7617_{\pm0.0004}}$ | $\mathbf{0.7631_{\pm0.0010}}$ | $\mathbf{0.7629_{\pm0.0006}}$ | $\mathbf{0.7629_{\pm0.0007}}$ | $\mathbf{0.7629_{\pm0.0002}}$ | $\mathbf{0.7629_{\pm0.0007}}$ | $\mathbf{0.7627_{\pm0.0012}}$ | $\mathbf{0.7626_{\pm0.0005}}$ |
| LZD | Adam | $0.7118_{\pm0.0030}$ | $0.6613_{\pm0.0060}$ | $0.6252_{\pm0.0011}$ | $0.6065_{\pm0.0034}$ | $0.7155_{\pm0.0018}$ | $0.6726_{\pm0.0011}$ | $0.6308_{\pm0.0014}$ | $0.6079_{\pm0.0068}$ |
| | MEDA | $0.7118_{\pm0.0030}$ | $0.7105_{\pm0.0052}$ | $0.7081_{\pm0.0033}$ | $0.7176_{\pm0.0010}$ | $0.7155_{\pm0.0018}$ | $0.7162_{\pm0.0021}$ | $0.7177_{\pm0.0007}$ | $0.7162_{\pm0.0005}$ |
| | SAM | $0.7130_{\pm0.0028}$ | $0.6696_{\pm0.0024}$ | $0.6341_{\pm0.0033}$ | $0.6155_{\pm0.0067}$ | $0.7161_{\pm0.0019}$ | $0.6795_{\pm0.0042}$ | $0.6360_{\pm0.0024}$ | $0.6111_{\pm0.0023}$ |
| | AdamW | $0.7132_{\pm0.0015}$ | $0.7135_{\pm0.0008}$ | $0.7140_{\pm0.0029}$ | $0.7139_{\pm0.0006}$ | $0.7143_{\pm0.0004}$ | $0.7138_{\pm0.0016}$ | $0.7142_{\pm0.0009}$ | $0.7131_{\pm0.0002}$ |
| | AdamAR | $\mathbf{0.7229_{\pm0.0008}}$ | $\mathbf{0.7235_{\pm0.0022}}$ | $\mathbf{0.7241_{\pm0.0010}}$ | $\mathbf{0.7234_{\pm0.0012}}$ | $\mathbf{0.7233_{\pm0.0008}}$ | $\mathbf{0.7240_{\pm0.0007}}$ | $\mathbf{0.7246_{\pm0.0013}}$ | $\mathbf{0.7240_{\pm0.0010}}$ |

Table 9: Comparison of average test AUC with xDeepFM and WuKong models using Adam optimizer.

| Dataset | Method | xDeepFM | | | | WuKong | | | |
|---|---|---|---|---|---|---|---|---|---|
| | | E1 | E2 | E3 | E4 | E1 | E2 | E3 | E4 |
| iPinYou | Adam | $0.7590_{\pm0.0026}$ | $0.7391_{\pm0.0017}$ | $0.6969_{\pm0.0081}$ | $0.6844_{\pm0.0083}$ | $0.7611_{\pm0.0035}$ | $0.7442_{\pm0.0079}$ | $0.7082_{\pm0.0075}$ | $0.6915_{\pm0.0035}$ |
| | MEDA | $0.7590_{\pm0.0026}$ | $0.7584_{\pm0.0014}$ | $0.7575_{\pm0.0027}$ | $0.7597_{\pm0.0005}$ | $0.7611_{\pm0.0035}$ | $0.7663_{\pm0.0010}$ | $0.7662_{\pm0.0021}$ | $0.7706_{\pm0.0028}$ |
| | SAM | $0.7565_{\pm0.0091}$ | $0.7517_{\pm0.0019}$ | $0.7413_{\pm0.0010}$ | $0.7274_{\pm0.0007}$ | $0.7033_{\pm0.0301}$ | $0.7485_{\pm0.0187}$ | $0.7465_{\pm0.0084}$ | $0.7306_{\pm0.0185}$ |
| | AdamW | $0.7607_{\pm0.0043}$ | $0.7660_{\pm0.0019}$ | $0.7651_{\pm0.0022}$ | $0.7574_{\pm0.0028}$ | $0.7579_{\pm0.0012}$ | $0.7634_{\pm0.0056}$ | $0.7605_{\pm0.0018}$ | $0.7511_{\pm0.0086}$ |
| | AdamAR | $\mathbf{0.7725_{\pm0.0026}}$ | $\mathbf{0.7733_{\pm0.0008}}$ | $\mathbf{0.7711_{\pm0.0003}}$ | $\mathbf{0.7678_{\pm0.0015}}$ | $\mathbf{0.7653_{\pm0.0021}}$ | $\mathbf{0.7736_{\pm0.0029}}$ | $\mathbf{0.7748_{\pm0.0025}}$ | $\mathbf{0.7736_{\pm0.0027}}$ |
| Amazon | Adam | $0.8460_{\pm0.0010}$ | $0.8535_{\pm0.0019}$ | $0.8287_{\pm0.0001}$ | $0.8163_{\pm0.0028}$ | $0.8580_{\pm0.0008}$ | $0.8600_{\pm0.0022}$ | $0.8348_{\pm0.0027}$ | $0.8232_{\pm0.0039}$ |
| | MEDA | $0.8460_{\pm0.0010}$ | $0.8519_{\pm0.0009}$ | $0.8551_{\pm0.0017}$ | $0.8562_{\pm0.0004}$ | $0.8580_{\pm0.0008}$ | $0.8611_{\pm0.0031}$ | $0.8621_{\pm0.0018}$ | $0.8626_{\pm0.0005}$ |
| | SAM | $\mathbf{0.8505_{\pm0.0020}}$ | $0.8587_{\pm0.0017}$ | $0.8390_{\pm0.0010}$ | $0.8232_{\pm0.0018}$ | $\mathbf{0.8588_{\pm0.0011}}$ | $0.8639_{\pm0.0005}$ | $0.8404_{\pm0.0062}$ | $0.8225_{\pm0.0062}$ |
| | AdamW | $0.8446_{\pm0.0011}$ | $0.8557_{\pm0.0020}$ | $0.8381_{\pm0.0014}$ | $0.8240_{\pm0.0013}$ | $0.8564_{\pm0.0017}$ | $0.8632_{\pm0.0015}$ | $0.8478_{\pm0.0047}$ | $0.8349_{\pm0.0051}$ |
| | AdamAR | $0.8483_{\pm0.0004}$ | $\mathbf{0.8676_{\pm0.0019}}$ | $\mathbf{0.8687_{\pm0.0015}}$ | $\mathbf{0.8675_{\pm0.0011}}$ | $0.8582_{\pm0.0008}$ | $\mathbf{0.8696_{\pm0.0014}}$ | $\mathbf{0.8693_{\pm0.0010}}$ | $\mathbf{0.8664_{\pm0.0013}}$ |
| Avazu | Adam | $0.7488_{\pm0.0007}$ | $0.7217_{\pm0.0038}$ | $0.7019_{\pm0.0052}$ | $0.6869_{\pm0.0045}$ | $0.7514_{\pm0.0037}$ | $0.7360_{\pm0.0048}$ | $0.7141_{\pm0.0126}$ | $0.7079_{\pm0.0076}$ |
| | MEDA | $0.7488_{\pm0.0007}$ | $0.7489_{\pm0.0028}$ | $0.7505_{\pm0.0006}$ | $0.7506_{\pm0.0016}$ | $0.7514_{\pm0.0037}$ | $0.7548_{\pm0.0014}$ | $0.7553_{\pm0.0004}$ | $0.7571_{\pm0.0013}$ |
| | SAM | $0.7484_{\pm0.0017}$ | $0.7190_{\pm0.0065}$ | $0.7010_{\pm0.0046}$ | $0.6902_{\pm0.0017}$ | $0.7513_{\pm0.0022}$ | $0.7333_{\pm0.0039}$ | $0.7155_{\pm0.0038}$ | $0.7156_{\pm0.0036}$ |
| | AdamW | $0.7583_{\pm0.0015}$ | $0.7582_{\pm0.0006}$ | $0.7589_{\pm0.0010}$ | $0.7593_{\pm0.0015}$ | $0.7542_{\pm0.0029}$ | $0.7547_{\pm0.0019}$ | $0.7558_{\pm0.0006}$ | $0.7564_{\pm0.0022}$ |
| | AdamAR | $\mathbf{0.7628_{\pm0.0008}}$ | $\mathbf{0.7633_{\pm0.0006}}$ | $\mathbf{0.7638_{\pm0.0005}}$ | $\mathbf{0.7636_{\pm0.0006}}$ | $\mathbf{0.7624_{\pm0.0005}}$ | $\mathbf{0.7612_{\pm0.0005}}$ | $\mathbf{0.7623_{\pm0.0008}}$ | $\mathbf{0.7624_{\pm0.0011}}$ |
| LZD | Adam | $0.7164_{\pm0.0017}$ | $0.6787_{\pm0.0025}$ | $0.6321_{\pm0.0050}$ | $0.6050_{\pm0.0096}$ | $0.7101_{\pm0.0054}$ | $0.6645_{\pm0.0112}$ | $0.6102_{\pm0.0074}$ | $0.6044_{\pm0.0035}$ |
| | MEDA | $0.7164_{\pm0.0017}$ | $0.7170_{\pm0.0033}$ | $0.7152_{\pm0.0011}$ | $0.7139_{\pm0.0050}$ | $0.7101_{\pm0.0054}$ | $0.7170_{\pm0.0010}$ | $0.7129_{\pm0.0046}$ | $0.7128_{\pm0.0067}$ |
| | SAM | $0.7166_{\pm0.0023}$ | $0.6750_{\pm0.0083}$ | $0.6344_{\pm0.0097}$ | $0.6134_{\pm0.0070}$ | $0.7146_{\pm0.0038}$ | $0.6713_{\pm0.0034}$ | $0.6443_{\pm0.0062}$ | $0.6212_{\pm0.0120}$ |
| | AdamW | $0.7142_{\pm0.0016}$ | $0.7151_{\pm0.0006}$ | $0.7139_{\pm0.0032}$ | $0.7139_{\pm0.0005}$ | $0.7115_{\pm0.0022}$ | $0.7135_{\pm0.0013}$ | $0.7152_{\pm0.0015}$ | $0.7142_{\pm0.0029}$ |
| | AdamAR | $\mathbf{0.7244_{\pm0.0008}}$ | $\mathbf{0.7256_{\pm0.0003}}$ | $\mathbf{0.7242_{\pm0.0013}}$ | $\mathbf{0.7238_{\pm0.0009}}$ | $\mathbf{0.7227_{\pm0.0010}}$ | $\mathbf{0.7215_{\pm0.0023}}$ | $\mathbf{0.7208_{\pm0.0006}}$ | $\mathbf{0.7202_{\pm0.0019}}$ |

Table 10: Comparison of average test AUC with DNN and WDL models using Adagrad optimizer.

| Dataset | Method | DNN | | | | WDL | | | |
|---|---|---|---|---|---|---|---|---|---|
| | | E1 | E2 | E3 | E4 | E1 | E2 | E3 | E4 |
| iPinYou | Adagrad | $0.7593_{\pm0.0032}$ | $0.6507_{\pm0.0040}$ | $0.6231_{\pm0.0183}$ | $0.6095_{\pm0.0054}$ | $0.7646_{\pm0.0021}$ | $0.6653_{\pm0.0116}$ | $0.6321_{\pm0.0092}$ | $0.6241_{\pm0.0107}$ |
| | MEDA | $0.7593_{\pm0.0032}$ | $0.7686_{\pm0.0026}$ | $0.7710_{\pm0.0020}$ | $0.7729_{\pm0.0035}$ | $0.7646_{\pm0.0021}$ | $0.7681_{\pm0.0053}$ | $0.7685_{\pm0.0032}$ | $0.7715_{\pm0.0013}$ |
| | SAM | $0.7576_{\pm0.0052}$ | $0.7661_{\pm0.0007}$ | $0.7487_{\pm0.0036}$ | $0.7303_{\pm0.0054}$ | $0.7624_{\pm0.0028}$ | $0.7676_{\pm0.0028}$ | $0.7518_{\pm0.0022}$ | $0.7369_{\pm0.0038}$ |
| | AdagradW | $0.7558_{\pm0.0029}$ | $0.7651_{\pm0.0034}$ | $0.7667_{\pm0.0014}$ | $0.7595_{\pm0.0050}$ | $0.7593_{\pm0.0051}$ | $0.7675_{\pm0.0020}$ | $0.7635_{\pm0.0038}$ | $0.7593_{\pm0.0006}$ |
| | AdagradAR | $\mathbf{0.7681_{\pm0.0034}}$ | $\mathbf{0.7754_{\pm0.0042}}$ | $\mathbf{0.7760_{\pm0.0001}}$ | $\mathbf{0.7744_{\pm0.0033}}$ | $\mathbf{0.7731_{\pm0.0022}}$ | $\mathbf{0.7772_{\pm0.0010}}$ | $\mathbf{0.7748_{\pm0.0016}}$ | $\mathbf{0.7720_{\pm0.0012}}$ |
| Amazon | Adagrad | $0.8438_{\pm0.0004}$ | $0.8402_{\pm0.0015}$ | $0.8141_{\pm0.0013}$ | $0.8042_{\pm0.0035}$ | $0.8406_{\pm0.0019}$ | $0.8345_{\pm0.0012}$ | $0.8085_{\pm0.0024}$ | $0.7982_{\pm0.0010}$ |
| | MEDA | $0.8438_{\pm0.0004}$ | $0.8481_{\pm0.0013}$ | $0.8495_{\pm0.0018}$ | $0.8505_{\pm0.0018}$ | $0.8406_{\pm0.0019}$ | $0.8470_{\pm0.0012}$ | $0.8497_{\pm0.0015}$ | $0.8510_{\pm0.0018}$ |
| | SAM | $\mathbf{0.8500_{\pm0.0017}}$ | $0.8527_{\pm0.0018}$ | $0.8361_{\pm0.0005}$ | $0.8193_{\pm0.0019}$ | $\mathbf{0.8490_{\pm0.0008}}$ | $0.7995_{\pm0.0905}$ | $0.8325_{\pm0.0031}$ | $0.8155_{\pm0.0017}$ |
| | AdagradW | $0.8428_{\pm0.0003}$ | $0.8578_{\pm0.0016}$ | $0.8563_{\pm0.0009}$ | $0.8535_{\pm0.0003}$ | $0.8424_{\pm0.0009}$ | $0.8553_{\pm0.0009}$ | $0.8531_{\pm0.0006}$ | $0.8498_{\pm0.0010}$ |
| | AdagradAR | $0.8479_{\pm0.0006}$ | $\mathbf{0.8659_{\pm0.0013}}$ | $\mathbf{0.8711_{\pm0.0002}}$ | $\mathbf{0.8712_{\pm0.0007}}$ | $0.8453_{\pm0.0008}$ | $\mathbf{0.8631_{\pm0.0009}}$ | $\mathbf{0.8687_{\pm0.0015}}$ | $\mathbf{0.8707_{\pm0.0009}}$ |
| Avazu | Adagrad | $0.7541_{\pm0.0012}$ | $0.7323_{\pm0.0016}$ | $0.7164_{\pm0.0010}$ | $0.7074_{\pm0.0011}$ | $0.7543_{\pm0.0013}$ | $0.7305_{\pm0.0019}$ | $0.7158_{\pm0.0026}$ | $0.7073_{\pm0.0016}$ |
| | MEDA | $0.7541_{\pm0.0012}$ | $0.7549_{\pm0.0007}$ | $0.7544_{\pm0.0015}$ | $0.7545_{\pm0.0010}$ | $0.7543_{\pm0.0013}$ | $0.7547_{\pm0.0012}$ | $0.7540_{\pm0.0009}$ | $0.7551_{\pm0.0007}$ |
| | SAM | $0.7553_{\pm0.0006}$ | $0.7333_{\pm0.0010}$ | $0.7199_{\pm0.0019}$ | $0.7094_{\pm0.0016}$ | $0.7557_{\pm0.0013}$ | $0.7328_{\pm0.0008}$ | $0.7178_{\pm0.0003}$ | $0.7079_{\pm0.0003}$ |
| | AdagradW | $0.7578_{\pm0.0005}$ | $0.7580_{\pm0.0006}$ | $0.7575_{\pm0.0005}$ | $0.7566_{\pm0.0016}$ | $0.7584_{\pm0.0005}$ | $0.7581_{\pm0.0014}$ | $0.7567_{\pm0.0014}$ | $0.7585_{\pm0.0010}$ |
| | AdagradAR | $\mathbf{0.7628_{\pm0.0005}}$ | $\mathbf{0.7629_{\pm0.0003}}$ | $\mathbf{0.7630_{\pm0.0009}}$ | $\mathbf{0.7624_{\pm0.0005}}$ | $\mathbf{0.7631_{\pm0.0006}}$ | $\mathbf{0.7634_{\pm0.0008}}$ | $\mathbf{0.7623_{\pm0.0009}}$ | $\mathbf{0.7626_{\pm0.0008}}$ |
| LZD | Adagrad | $0.7226_{\pm0.0018}$ | $0.6658_{\pm0.0018}$ | $0.6433_{\pm0.0014}$ | $0.6376_{\pm0.0038}$ | $0.7244_{\pm0.0007}$ | $0.6695_{\pm0.0001}$ | $0.6389_{\pm0.0017}$ | $0.6386_{\pm0.0001}$ |
| | MEDA | $0.7226_{\pm0.0018}$ | $0.7183_{\pm0.0025}$ | $0.7150_{\pm0.0014}$ | $0.7236_{\pm0.0036}$ | $0.7244_{\pm0.0007}$ | $0.7241_{\pm0.0016}$ | $0.7239_{\pm0.0013}$ | $0.7219_{\pm0.0025}$ |
| | SAM | $0.7213_{\pm0.0011}$ | $0.6761_{\pm0.0048}$ | $0.6446_{\pm0.0105}$ | $0.6222_{\pm0.0092}$ | $0.7229_{\pm0.0015}$ | $0.6839_{\pm0.0053}$ | $0.6481_{\pm0.0058}$ | $0.6266_{\pm0.0026}$ |
| | AdagradW | $0.7145_{\pm0.0017}$ | $0.7132_{\pm0.0019}$ | $0.7135_{\pm0.0019}$ | $0.7154_{\pm0.0012}$ | $0.7138_{\pm0.0022}$ | $0.7149_{\pm0.0006}$ | $0.7137_{\pm0.0012}$ | $0.7145_{\pm0.0012}$ |
| | AdagradAR | $\mathbf{0.7253_{\pm0.0022}}$ | $\mathbf{0.7247_{\pm0.0014}}$ | $\mathbf{0.7234_{\pm0.0020}}$ | $\mathbf{0.7241_{\pm0.0013}}$ | $\mathbf{0.7269_{\pm0.0001}}$ | $\mathbf{0.7258_{\pm0.0013}}$ | $\mathbf{0.7251_{\pm0.0023}}$ | $\mathbf{0.7257_{\pm0.0010}}$ |

Table 11: Comparison of average test AUC with xDeepFM and WuKong models using Adagrad optimizer.

| Dataset | Method | xDeepFM | | | | WuKong | | | |
|---|---|---|---|---|---|---|---|---|---|
| | | E1 | E2 | E3 | E4 | E1 | E2 | E3 | E4 |
| iPinYou | Adagrad | $0.7674_{\pm0.0016}$ | $0.6843_{\pm0.0083}$ | $0.6505_{\pm0.0187}$ | $0.6527_{\pm0.0032}$ | $0.7661_{\pm0.0012}$ | $0.6900_{\pm0.0071}$ | $0.6497_{\pm0.0022}$ | $0.6320_{\pm0.0290}$ |
| | MEDA | $0.7674_{\pm0.0016}$ | $0.7722_{\pm0.0014}$ | $0.7728_{\pm0.0019}$ | $0.7740_{\pm0.0015}$ | $0.7661_{\pm0.0012}$ | $0.7705_{\pm0.0050}$ | $0.7729_{\pm0.0013}$ | $0.7695_{\pm0.0016}$ |
| | SAM | $0.7637_{\pm0.0022}$ | $0.7661_{\pm0.0054}$ | $0.7499_{\pm0.0043}$ | $0.7363_{\pm0.0017}$ | $0.7409_{\pm0.0081}$ | $0.7678_{\pm0.0015}$ | $0.7525_{\pm0.0029}$ | $0.7427_{\pm0.0032}$ |
| | AdagradW | $0.7665_{\pm0.0002}$ | $0.7673_{\pm0.0025}$ | $0.7666_{\pm0.0018}$ | $0.7652_{\pm0.0008}$ | $0.7578_{\pm0.0040}$ | $0.7661_{\pm0.0047}$ | $0.7649_{\pm0.0012}$ | $0.7600_{\pm0.0061}$ |
| | AdagradAR | $\mathbf{0.7760_{\pm0.0025}}$ | $\mathbf{0.7768_{\pm0.0023}}$ | $\mathbf{0.7762_{\pm0.0019}}$ | $\mathbf{0.7745_{\pm0.0005}}$ | $\mathbf{0.7718_{\pm0.0025}}$ | $\mathbf{0.7776_{\pm0.0027}}$ | $\mathbf{0.7782_{\pm0.0036}}$ | $\mathbf{0.7774_{\pm0.0029}}$ |
| Amazon | Adagrad | $0.8405_{\pm0.0014}$ | $0.8374_{\pm0.0016}$ | $0.8103_{\pm0.0018}$ | $0.7996_{\pm0.0034}$ | $0.8491_{\pm0.0022}$ | $0.8472_{\pm0.0020}$ | $0.8156_{\pm0.0050}$ | $0.8046_{\pm0.0039}$ |
| | MEDA | $0.8405_{\pm0.0014}$ | $0.8463_{\pm0.0023}$ | $0.8476_{\pm0.0015}$ | $0.8500_{\pm0.0001}$ | $0.8491_{\pm0.0022}$ | $0.8569_{\pm0.0011}$ | $0.8587_{\pm0.0018}$ | $0.8609_{\pm0.0010}$ |
| | SAM | $\mathbf{0.8483_{\pm0.0003}}$ | $0.8521_{\pm0.0005}$ | $0.8325_{\pm0.0009}$ | $0.8170_{\pm0.0024}$ | $\mathbf{0.8556_{\pm0.0029}}$ | $0.8567_{\pm0.0029}$ | $0.8356_{\pm0.0058}$ | $0.8174_{\pm0.0084}$ |
| | AdagradW | $0.8410_{\pm0.0013}$ | $0.8565_{\pm0.0011}$ | $0.8522_{\pm0.0009}$ | $0.8508_{\pm0.0014}$ | $0.8513_{\pm0.0019}$ | $0.8630_{\pm0.0015}$ | $0.8617_{\pm0.0025}$ | $0.8606_{\pm0.0031}$ |
| | AdagradAR | $0.8444_{\pm0.0014}$ | $\mathbf{0.8641_{\pm0.0008}}$ | $\mathbf{0.8681_{\pm0.0007}}$ | $\mathbf{0.8703_{\pm0.0004}}$ | $0.8538_{\pm0.0010}$ | $\mathbf{0.8690_{\pm0.0018}}$ | $\mathbf{0.8708_{\pm0.0005}}$ | $\mathbf{0.8700_{\pm0.0007}}$ |
| Avazu | Adagrad | $0.7550_{\pm0.0002}$ | $0.7319_{\pm0.0021}$ | $0.7160_{\pm0.0006}$ | $0.7069_{\pm0.0011}$ | $0.7548_{\pm0.0015}$ | $0.7311_{\pm0.0024}$ | $0.7160_{\pm0.0024}$ | $0.7041_{\pm0.0059}$ |
| | MEDA | $0.7550_{\pm0.0002}$ | $0.7538_{\pm0.0032}$ | $0.7550_{\pm0.0009}$ | $0.7553_{\pm0.0003}$ | $0.7548_{\pm0.0015}$ | $0.7551_{\pm0.0008}$ | $0.7556_{\pm0.0012}$ | $0.7564_{\pm0.0015}$ |
| | SAM | $0.7564_{\pm0.0007}$ | $0.7332_{\pm0.0021}$ | $0.7198_{\pm0.0023}$ | $0.7094_{\pm0.0007}$ | $0.7558_{\pm0.0021}$ | $0.7353_{\pm0.0016}$ | $0.7211_{\pm0.0027}$ | $0.7110_{\pm0.0033}$ |
| | AdagradW | $0.7583_{\pm0.0013}$ | $0.7581_{\pm0.0020}$ | $0.7580_{\pm0.0004}$ | $0.7585_{\pm0.0009}$ | $0.7524_{\pm0.0038}$ | $0.7555_{\pm0.0004}$ | $0.7555_{\pm0.0005}$ | $0.7563_{\pm0.0011}$ |
| | AdagradAR | $\mathbf{0.7636_{\pm0.0001}}$ | $\mathbf{0.7633_{\pm0.0013}}$ | $\mathbf{0.7635_{\pm0.0008}}$ | $\mathbf{0.7633_{\pm0.0007}}$ | $\mathbf{0.7628_{\pm0.0006}}$ | $\mathbf{0.7626_{\pm0.0008}}$ | $\mathbf{0.7627_{\pm0.0006}}$ | $\mathbf{0.7620_{\pm0.0007}}$ |
| LZD | Adagrad | $0.7222_{\pm0.0003}$ | $0.6675_{\pm0.0022}$ | $0.6400_{\pm0.0044}$ | $0.6382_{\pm0.0031}$ | $0.7247_{\pm0.0047}$ | $0.6726_{\pm0.0038}$ | $0.6412_{\pm0.0052}$ | $0.6339_{\pm0.0036}$ |
| | MEDA | $0.7222_{\pm0.0003}$ | $0.7237_{\pm0.0028}$ | $0.7253_{\pm0.0022}$ | $0.7256_{\pm0.0006}$ | $0.7247_{\pm0.0047}$ | $0.7237_{\pm0.0024}$ | $0.7234_{\pm0.0026}$ | $0.7239_{\pm0.0036}$ |
| | SAM | $0.7229_{\pm0.0019}$ | $0.6854_{\pm0.0052}$ | $0.6480_{\pm0.0066}$ | $0.6268_{\pm0.0041}$ | $0.7248_{\pm0.0018}$ | $0.6802_{\pm0.0094}$ | $0.6479_{\pm0.0184}$ | $0.6299_{\pm0.0195}$ |
| | AdagradW | $0.7155_{\pm0.0008}$ | $0.7135_{\pm0.0004}$ | $0.7117_{\pm0.0035}$ | $0.7150_{\pm0.0007}$ | $0.7127_{\pm0.0005}$ | $0.7111_{\pm0.0012}$ | $0.7147_{\pm0.0003}$ | $0.7136_{\pm0.0012}$ |
| | AdagradAR | $\mathbf{0.7273_{\pm0.0004}}$ | $\mathbf{0.7268_{\pm0.0016}}$ | $\mathbf{0.7264_{\pm0.0016}}$ | $\mathbf{0.7263_{\pm0.0011}}$ | $\mathbf{0.7269_{\pm0.0026}}$ | $\mathbf{0.7260_{\pm0.0003}}$ | $\mathbf{0.7252_{\pm0.0001}}$ | $\mathbf{0.7239_{\pm0.0029}}$ |

