# OpenReview forum: "Adaptive Regularization for Large-Scale Sparse Feature Embedding Models"
_ICLR.cc/2026/Conference — ICLR 2026 Poster_

### Official Review · Reviewer_oGeC · 2025-10-30

**Soundness:** 3
**Presentation:** 3
**Contribution:** 3
**Rating:** 4
**Confidence:** 4

**Summary:**

The authors study the one-epoch overfitting phenomenon in pCTR and pCVR models. They first conduct a theoretical analysis using Rademacher complexity, indicating that the norm of sparse embeddings leads to poor generalization. The authors propose an adaptive regularization method (AdamAR / AdagradAR)  with a dynamic strength of the embedding regularization terms. Experiments on several public datasets demonstrate the effectiveness of the proposed method.

**Strengths:**

1. Motivation. The paper studies a well-known and critical problem in pCXR prediction - the "one-epoch" overfitting problem.

2. Theoretical Grounding. The analysis links overfitting to Rademacher complexity, showing that embedding norm growth drives generalization error, giving a principled basis for adaptive regularization.

3. Technical Novelty. The proposed method is well-derived and elegant, with a limited impact on the model training.

**Weaknesses:**

1. Extension to Explicit Interaction Models. The analysis is only in DNNs, whose effectiveness is limited. It would be great to extend the theoretical discussion to explicit interaction models such as FM or CrossNet in DCN V2.

2. Experiments. Besides Wukong (BTW, the citation of the Wukong paper is wrong), other models in the experiment section are a bit out of date. It would be great to add experiments on the new SOTAs, e.g., DCN V2, xDeepFM.

3. Writing. The abstract contains limited concrete content. It would be great to be a bit more specific on the proposed theoretical analysis and method.

**Questions:**

1. Can the complexity analysis be extended to explicit feature interaction models like FM? It would be great to include such analysis since most modern recommendation systems heavily rely on explicit interactions.

2. How does the rewriting from Eq. 5 to Eq. 6 happen?

3. I'm curious about the effect of the proposed regularization method on the dimensional robustness of the learned embeddings. There are several recent works discussing the dimensional collapse (DC) issue in recommendation models [1,2], and I wonder whether there is a connection between the one-epoch phenomenon and DC. Can you provide an analysis of the singular spectrum of sparse embeddings?

[1]. On the embedding collapse when scaling up recommendation models. ICML 2024.

[2]. Balancing Embedding Spectrum for Recommendation. 2025.

[3]. Towards Mitigating Dimensional Collapse of Representations in Collaborative Filtering. WSDM 2024.

---

> ### Author Response · Authors · 2025-11-21
>
> We thank the reviewer for their valuable suggestions and insightful intuition, and present our detailed answer and analysis below.
>
> Q1: Extension to Explicit Interaction Models. The analysis is only in DNNs, whose effectiveness is limited. It would be great to extend the theoretical discussion to explicit interaction models such as FM or CrossNet in DCN V2.
>
> A1: We provide a detailed discussion of the Rademacher complexity bound for FM in appendix D, which demonstrates that the embedding parameters also dominate the bound. For other types of models, we may write a separate paper to further explore the Rademacher complexity bound, as the current work primarily focuses on addressing the one-epoch issue.
>
> Q2: Experiments. Besides Wukong (BTW, the citation of the Wukong paper is wrong), other models in the experiment section are a bit out of date. It would be great to add experiments on the new SOTAs, e.g., DCN V2, xDeepFM.
>
> A2: We apologize for the incorrect citation and have corrected the reference to the WuKong paper. In the experimental section, we replaced the DeepFM model with xDeepFM and found that our method continues to perform well with the xDeepFM backbone. We also conducted experiments using the DCNv2 backbone, and the results were similar to those obtained with xDeepFM. Due to space limitations, we only present the xDeepFM results in the paper. However, the DCNv2 experiment results can be found in our open-source anonymous code on GitHub.
>
> Q3: Writing. The abstract contains limited concrete content. It would be great to be a bit more specific on the proposed theoretical analysis and method.
>
> A3: Thank you for the kind reminder. We have rewritten the abstract to provide more details on the analysis method and our proposed approach.
>
> Q4: Can the complexity analysis be extended to explicit feature interaction models like FM? It would be great to include such analysis since most modern recommendation systems heavily rely on explicit interactions.
>
> A4: We provide it in appendix D equation 33.
>
> Q5: How does the rewriting from Eq. 5 to Eq. 6 happen?
>
> A5: Since each $\\mathbf x_i(t)$ is a one-hot vector, it follows that $\forall i, \|| \\mathbf x_{i}(t) \||^2 = 1$. Therefore, $\sqrt{ \sum_{t=1}^T \sum_{i=1}^S \|| \\mathbf x_{i}(t) \||^2 } = \sqrt{ST}$. By substituting this result into equation 5, we obtain equation 6. We have reformulated equation 6 and highlighted the condition $\||\\mathbf x_{i}(t)\||^2=1$ in section 2.2 to make the derivation easier to follow.
>
> Q6: I'm curious about the effect of the proposed regularization method on the dimensional robustness of the learned embeddings. There are several recent works discussing the dimensional collapse (DC) issue in recommendation models [1,2], and I wonder whether there is a connection between the one-epoch phenomenon and DC. Can you provide an analysis of the singular spectrum of sparse embeddings?
>
> A6: We have read [1] carefully, and the paper points out that naively enlarging the embedding size does not necessarily yield significant performance gains due to the feature interaction module of models. This is indeed a valuable insight that we have also observed in our production environment. However, our work primarily addresses the one-epoch issue, which can occur even when the embedding size is small. To demonstrate robustness across different embedding sizes, we present additional experiments in appendix E. The singular spectrum of the embedding feature "IP" over four epochs can also be found in appendix E. It illustrates a substantial increase in the sum of singular values for the Adam and SAM methods, whereas our method remains stable. In addition, the information abundance (IA) defined in [1] can be preserved by our method even after multi‑epoch training, reflecting a balanced distribution in the embedding vector space. In contrast, all other methods except MEDA exhibit a deterioration in the IA metric.

---

> > ### Comment · Reviewer_oGeC · 2025-11-28
> >
> > Thanks for the detailed responses, which resolve most of my concerns. I will raise my rating score.

---

> > > ### Author Response · Authors · 2025-11-28
> > >
> > > Thank you for your valuable suggestions and questions, as well as for your recognition of our work.

---

> ### Comment · Area_Chair_nunP · 2025-11-26
>
> Dear colleagues,
>
> The authors have provided their rebuttal. Can you review it and confirm if there are any further concerns about the work?
>
> AC

---

### Official Review · Reviewer_fK43 · 2025-10-30

**Soundness:** 3
**Presentation:** 2
**Contribution:** 2
**Rating:** 6
**Confidence:** 4

**Summary:**

This paper investigates the prevalent phenomenon of "one-epoch overfitting" in large-scale sparse feature embedding models, especially pertinent to click-through rate and conversion rate estimation in online advertising and recommendation systems. The authors provide a theoretical analysis rooted in Rademacher complexity, pinpointing the growth of embedding norms as the root cause of deteriorating generalization when training persists beyond a single epoch. To address this, they propose an adaptive regularization technique that dynamically adjusts regularization strength per embedding based on occurrence intervals, integrating seamlessly with standard optimizers like Adam and Adagrad. Comprehensive experiments across public and industrial datasets demonstrate that the proposed method outperforms existing baselines in mitigating overfitting and promoting stable, improved performance during multi-epoch training.

**Strengths:**

* Theoretical Insights: The paper offers a coherent theoretical framework using Rademacher complexity to explain one-epoch overfitting in sparse embeddings, clarifying a widely observed yet under-theorized phenomenon in deep CTR/CVR models (see Section 2.2 and 3).
* Principled Adaptive Method: Instead of relying on heuristics, the proposed adaptive regularization derives directly from the theoretical analysis, offering an intuitive and practical approach that is compatible with common optimizers (Section 3.1, Algorithm 1).
* Comprehensive Empirical Evaluation: The method is validated across a diverse set of datasets (iPinYou, Amazon, Avazu, proprietary LZD) and backbone architectures (DNN, WDL, DeepFM, WuKong), demonstrating strong and consistent improvements over multiple baselines (Tables 1, 2).

**Weaknesses:**

* Limited Positioning Relative to Broader Regularization Literature: While the paper covers baseline regularization approaches (L1, L2, weight decay), it does not sufficiently engage with recent or foundational work on sparsity-driven regularization at both the architectural and optimization levels. For instance, no discussion is provided relating their adaptive method to pruning/growth strategies or thresholding-based approaches, even though such techniques are central to the sparsity literature.
* Lack of Ablations: There is little to no ablation study dissecting critical components of the proposed method, such as the sensitivity to the base regularization coefficient $\alpha$ beyond Figure 3, or the explicit contribution of occurrence interval estimation. It remains unclear how much each modeling choice contributes to the performance gains.
* Hyperparameter Selection Transparency: Although hyperparameters are said to be chosen by grid search (Section 4.1.3), the details are sequestered in the appendix and visualization (Figure 3), limiting transparency and reproducibility. The main text should provide a concise summary of grid search procedures and best practice recommendations.
* Ambiguity in Some Notation and Algorithm Details: Certain notations, particularly around frequency and interval estimation for embeddings (Section 3.1), could be more precise. For instance, the paper uses $I_{ij}$ as both a stochastic occurrence interval and an empirical quantity with little justification for the estimation method used during optimizer steps.
* Insufficient Theoretical Analysis of Method Limitations: The theoretical section does not address possible drawbacks or edge cases (such as whether the adaptive regularization could lead to underfitting for extremely rare or never-seen values), nor the impact of frequency misestimation on convergence or generalization.
* Lack of Comparison to other LLM-based models: There are some new algorithms which are LLM-based recommendation algorithm, and this paper does not consider them.

**Questions:**

* How Sensitive is the Proposed Method to Hyperparameter $\alpha$ Across Datasets?
Figure 3 provides some evidence, but would the authors provide additional experimental data on extreme values of $\alpha$, and practical guidelines for selection especially in large-scale real-world datasets?
* Handling Never-seen or Extremely Rare Features:
Does the adaptive regularization approach systematically bias against tail classes? How does it perform in cold-start (new feature value) scenarios, and is there risk of underfitting for features with vanishing sample frequency? Can the method adapt dynamically if sparsity levels change after deployment?
* Ablation on Update Interval Estimation:
Would the authors run ablation studies to decouple the effect of interval-based regularization vs. embedding-based regularization overall?
* Comparison with Other Adaptive/Hybrid Regularization Approaches:
How does the proposed method perform compared to thresholding or pruning-based strategies, as well as non-uniform L1/L2 variants found in the sparse modeling literature? Would Table 1/2 results meaningfully shift under such baselines?
* Computational Overhead and Scalability:
Are there notable compute or memory implications to tracking per-embedding update intervals, especially as feature cardinality scales beyond the reported datasets?

---

> ### Author Response · Authors · 2025-11-21
>
> We thank the reviewer for their detailed questions and provide our responses below.
>
> Q1:Limited Positioning Relative to Broader Regularization Literature.
>
> A1:We include SAM as a benchmark as suggested by reviewer amMd. By explicitly penalizing the sharpness of the loss landscape during each training step, this method encourages convergence to flat minima, thereby consistently enhancing generalization on a wide range of tasks. We find that although it performs well on the Amazon dataset under single-epoch training, it cannot address the one-epoch issue.
>
> Q2:Hyperparameter Selection Transparency.
>
> A2:For transparency, we provide the detailed formulation of our search method in section 4.1, noting that the grid search over values of the form $10^{n}$ where $n$ ranges from $-6.5$ to $-1$ with a step size of $0.5$. In appendix I, we provide the grid search results across all datasets.
>
> Q3:Ambiguity in Some Notation and Algorithm Details.
>
> A3:We have changed the notation for the stochastic occurrence interval to $\\mathcal I_{ij}$ and slightly revised the empirical quantity at step $k$ to $I^k_{ij}$ to improve clarity for readers in section 3.1.
>
> Q4:Insufficient Theoretical Analysis of Method Limitations.
>
> A4:The convergence analysis is given in section 3.3 and appendix B.1. The bucket analysis in section 4.5 examines performance across different feature frequencies.
>
> Q5:Lack of Comparison to other LLM-based models.
>
> A5:LLM-based recommendation models have attracted attention in recent research and demonstrated some promising results. However, in industrial applications, mainstream CTR/CVR estimation still relies on models with large‑scale sparse embeddings, which differ substantially from LLM‑based recommendation models, whose parameters are primarily concentrated in the MLP‑like components. Our work mainly focuses on models in which the embedding parameters account for the majority of the total model size. While LLM-based recommendation models are certainly worthy of further investigation, we plan to explore them in a separate study in the future.
>
> Q6:How Sensitive is the Proposed Method to Hyperparameter  Across Datasets? Figure 3 provides some evidence, but would the authors provide additional experimental data on extreme values of , and practical guidelines for selection especially in large-scale real-world datasets?
>
> A6:We have added the grid search result across all datasets in appendix I. In large-scale real-world datasets such as ours, we use a two-week sample to perform a grid search and determine suitable hyperparameters, which are then applied for training on the complete dataset.
>
> Q7:Handling Never-seen or Extremely Rare Features?
>
> A7:In section 4.5, we conduct bucket analysis to evaluate performance across different users, since "IP" can be interpreted as a proxy for a user. The results show that our method performs well across all buckets and achieves particularly strong performance in the high‑frequency bucket due to the larger norm budget available. We think that infrequent users or items can be more effectively modeled using relatively dense features rather than sparse, potentially unreliable features. In real production environments, we perform daily incremental training to mitigate regularization lag under distribution shifts.
>
> Q8:Ablation on Update Interval Estimation?
>
> A8:We apologize for the previously unclear explanation of AdamW/AdagradW setting. In our experiments, weight decay is applied only to the embedding layers for AdamW and AdagradW, which may serve as an ablation comparison between embedding layer regularization and our proposed method. We have added this explanation in section 4.1. Furthermore, we present an ablation study across different buckets of feature frequency to clarify the effect of occurrence‑interval estimation in section 4.5.
>
>
> Q9:Computational Overhead and Scalability?
>
> A9: We discuss its impact in detail in section 3.1 and appendix F. The computational complexity still holds at $O(P)$, while the memory usage increases by approximately 33\% during the training stage. For industrial implementation, this method can be scaled further by deploying additional machines in the parameter server. Moreover, when multiple parameters share the same last valid update step (e.g., parameters in the same row of the embedding matrix), a single parameter can be used to record this step. This optimization can improve scalability and is already adopted in our production environment.

---

### Official Review · Reviewer_s7gA · 2025-10-31

**Soundness:** 4
**Presentation:** 3
**Contribution:** 3
**Rating:** 8
**Confidence:** 4

**Summary:**

The paper targets a common phenomenon in large-scale CTR/recommender models with highly sparse features: performance is often best after the first epoch, but further training leads to overfitting and degradation. The authors provide an explanation from the perspective of generalization bounds through Rademacher complexity as follows:

* most of the model capacity is concentrated in the huge embedding matrix, and in particular, in the very infrequent IDs’ embedding rows.
* the embedding rows with very infrequent IDs are updated on a much slower schedule than the dense part / MLP, so their norms can grow uncontrollably and thus inflate the overall capacity.

Based on this, the authors propose an adaptive regularization scheme that assigns stronger weight decay to infrequent embedding rows . They integrate this row-wise adaptive weight decay into Adam/Adagrad (termed AdamAR/AdagradAR), so that the sparser and less frequently updated a row is, the stronger the regularization it receives.

Experiments on multiple datasets and CTR architectures show that this approach can mitigate or even eliminate the “multi-epoch performance drop,” while slightly improving AUC.

**Strengths:**

* Problem importance. The paper tackles an industry-recognized issue in large-scale CTR models—a core component of nearly all advertising systems.
* Theoretical grounding. It links the empirical “train only one epoch” practice to a capacity-control perspective, giving the phenomenon a clearer theoretical justification.
* Practical simplicity. The proposed technique is easy to integrate: it only requires tracking the last valid update step for each embedding row and can be directly plugged into existing optimizers.

**Weaknesses:**

1. The connection to existing “frequency-aware” or “epoch-level reset” approaches (e.g., MEDA-style methods) is not elaborated systematically.
2. The experiments mainly report global metrics such as AUC/Logloss, but lack fine-grained bucket analyses (by feature frequency, long-tail IDs, cold-start features) to directly demonstrate that “low-frequency rows were actually controlled.”

**Questions:**

1. Does applying strong L2 to low-frequency rows in adamW reproduces the main gains? If so, that would validate the causal story more strongly.

---

> ### Author Response · Authors · 2025-11-21
>
> We appreciate your recognition of our work. In response to the issues you raised, we have conducted further investigation and provide our answers below.
>
> Q1:The connection to existing “frequency-aware” or “epoch-level reset” approaches (e.g., MEDA-style methods) is not elaborated systematically.
>
> A1:The connection between our method and MEDA is discussed in section 3.2, with equation 14 providing the exact formal relationship. Specifically, in MEDA, the embedding layers are reset only at epoch boundaries. In industrial practice, embedding layers are typically initialized to zero. Thus, at these boundaries (i.e., when $kB \bmod T = 0$), setting $I_p^k = 1/\alpha$ yields $\lambda_p = 1$, thereby resetting the embedding vectors. In contrast, $I_p^k = 0$ gives $\lambda_p = 0$, corresponding to the standard optimizer. So MEDA is a special case of our algorithm.
>
> Q2:The experiments mainly report global metrics such as AUC/Logloss, but lack fine-grained bucket analyses (by feature frequency, long-tail IDs, cold-start features) to directly demonstrate that "low-frequency rows were actually controlled."
>
> A2: We extend the bucket analysis in section 4.5 by reporting the regularization strength and test AUC for each bucket, grouped according to feature frequency. The results show that our method performs well across all buckets, and achieves particularly strong performance in the high‑frequency bucket due to the larger norm budget available. Figure 3(c) indicates that the low-frequency rows were effectively controlled.
>
> Q3:Does applying strong L2 to low-frequency rows in adamW reproduces the main gains? If so, that would validate the causal story more strongly.
>
> A:3 Thanks for the kind reminder of the causal problem. We conduct an ablation experiment in section 4.5 to clarify the effect of occurrence interval estimation. Table 3 shows that, compared with using a constant weight decay, decreasing the weight decay for high-frequency features and increasing it for low-frequency features can further improve performance while alleviating the one-epoch issue.

---

> ### Comment · Area_Chair_nunP · 2025-11-26
>
> Dear colleagues,
>
> The authors have provided their rebuttal. Can you review it and confirm if there are any further concerns about the work?
>
> AC

---

### Official Review · Reviewer_amMd · 2025-11-02

**Soundness:** 4
**Presentation:** 3
**Contribution:** 3
**Rating:** 6
**Confidence:** 3

**Summary:**

This paper analyzes the one-epoch overfitting phenomenon in large-scale CTR/CVR models with sparse embeddings. Using Rademacher complexity, the authors show that overfitting arises from unconstrained embedding norm growth for infrequent features. They propose an adaptive regularization method that dynamically adjusts regularization strength by feature update intervals, forming optimizers. Experiments on public and industrial datasets demonstrate consistent performance gains, improved generalization, and easy integration into existing training pipelines.

**Strengths:**

A substantive assessment of the strengths of the paper, touching on each of the following dimensions: originality, quality, clarity, and significance. We encourage reviewers to be broad in their definitions of originality and significance. For example, originality may arise from a new definition or problem formulation, creative combinations of existing ideas, application to a new domain, or removing limitations from prior results.

1. Sound and well-motivated analysis linking embedding sparsity and overfitting.
2. Adaptive rule is intuitive and easy to implement.
3. Multi-dataset, multi-optimizer validation with consistent gains.
4. Addresses a critical and under-theorized issue in real-world recommender systems.
5.Provides a clear explanation for why rare features require stronger regularization.

**Weaknesses:**

1. The analysis assumes i.i.d. sampling and bounded feature norms, which may not hold in practice for long-tail industrial data.
2. The derivation of adaptive coefficients relies on approximating feature frequency by occurrence intervals; this estimation may introduce stochastic noise not formally analyzed.
3. The differentiability of \phi(\tau_{ij}) and the KKT-based derivation rest on idealized smoothness assumptions that may not hold in real deep networks.
4. The paper doesn’t rigorously study how the adaptive coefficient \lambda_{ij} affects optimizer dynamics or convergence stability, especially in non-convex regimes.
5. A theoretical guarantee on convergence rate or bias induced by adaptive decay would strengthen the work.
6. Tracking the “last valid update step” for every embedding vector may introduce memory overhead in billion-scale models. The paper does not discuss memory/time complexity or optimization strategies for this in production.
7. While the baselines are relevant, comparisons with more modern adaptive regularization or implicit bias control methods  e.g., SAM, AdaNorm, or adaptive dropout  would enhance credibility.
8. No statistical significance testing is reported for AUC gains, though the improvements are small (≈0.002–0.01).
9. The paper briefly mentions online deployment but does not analyze potential trade-offs e.g., regularization lag under distribution shift, interpretability implications, or fairness across infrequent users/items.

**Questions:**

Please refer to the weaknesses.

---

> ### Author Response · Authors · 2025-11-21
>
> We are grateful for the reviewer’s valuable comment. We have carefully examined the identified weaknesses and present our detailed response below.
>
> Q1: The analysis assumes i.i.d. sampling and bounded feature norms, which may not hold in practice for long-tail industrial data.
>
> A1: In our real-world industrial environment, distribution shifts occur on a daily basis. In particular, on e-commerce platforms, promotional campaigns happen very frequently, which can violate the i.i.d. assumption. However, We introduce this assumption in our study for three main reasons:
>
> 1. We can perform daily incremental training to adapt to distribution shifts. Except during promotional campaign periods, our empirical tests show that the test dataset performance is essentially identical to the training dataset performance. This observation provides intuitive support for the i.i.d. assumption.
> 2. In many existing analyses of non-i.i.d. generalization error bounds, alternative assumptions are introduced. However, these assumptions also may fail to hold in our industrial settings and may lead to even more severe analytical limitations.
> 3. In the i.i.d. setting, well‑established and easy‑to‑use analytical tools are available for bounding the generalization error. These tools enable rapid verification of theoretical results against empirical observations in our production environment.
>
> Q2: The derivation of adaptive coefficients relies on approximating feature frequency by occurrence intervals; this estimation may introduce stochastic noise not formally analyzed.
>
> A2: We introduce proposition 3 in section 3.3 to analyze the impact of stochastic noise on the minimum convergence of the Adam optimizer. The result shows that such noise only affects the constant term in the bound given in equation 15. Furthermore, appendix B.2 discusses how equation 11 ensures that the noise remains bounded.
>
> Q3: The differentiability of $\varphi(\tau_{ij})$ and the KKT-based derivation rest on idealized smoothness assumptions that may not hold in real deep networks.
>
> A3: This is indeed a relatively strong assumption in practical deep networks. However, we address it under non-smooth conditions by employing the Clarke subdifferential, as shown in appendix C. By introducing a Lipschitz assumption, proposition 1 still holds. We adopt the smoothness assumption here primarily to present the result in a more direct and accessible manner for readers, and it would be easier to understand.
>
> Q4: The paper doesn't rigorously study how the adaptive coefficient $\lambda_{ij}$ affects optimizer dynamics or convergence stability especially in non-convex regimes. A theoretical guarantee on convergence rate or bias induced by adaptive decay would strengthen the work
>
> A4: We analyze the minimum convergence under the Lipschitz assumption in non-convex settings. Details are provided in section 3.3 and appendix B.1.
>
> Q5: The paper does not discuss memory/time complexity or optimization strategies for this in production.
>
> A5: We discuss its impact in detail in section 3.1 and appendix F. The computational complexity still holds at $O(P)$, while the memory usage increases by approximately 33\% during training stage if we store all parameters' last valid update step. Notably, for parameters belonging to the same row of an embedding matrix, this last valid update step can be shared, thereby reducing memory usage in real production environments.
>
> Q6: While the baselines are relevant, comparisons with more modern adaptive regularization or implicit bias control methods e.g., SAM, AdaNorm, or adaptive dropout would enhance credibility.
>
> A6: We appreciate the methods provided, which offer valuable insights. We include SAM as a benchmark and find that it cannot address the one-epoch issue, as its design is unrelated to the norm budget control of embedding layers. However, it performs well on the Amazon dataset under single-epoch training.
>
> Q7: No statistical significance testing is reported for AUC gains, though the improvements are small.
>
> A7: We repeat each experiment 3 times and report the average results in section 4.3 and give the standard deviation and detailed scalability of the results in appendix J.
>
> Q8: The paper briefly mentions online deployment but does not analyze potential trade-offs.
>
> A8: In section 4.5, we conduct bucket analysis to evaluate performance across different users, since "IP" can be interpreted as a proxy for a user. The results show that our method performs well across all buckets, and achieves particularly strong performance in the high‑frequency bucket due to the larger norm budget available. We think that infrequent users or items can be more effectively modeled using relatively dense features rather than sparse, potentially unreliable features. In addition, we perform daily incremental training of our models to mitigate regularization lag under distribution shifts.

---

> ### Comment · Area_Chair_nunP · 2025-11-26
>
> Dear colleagues,
>
> The authors have provided their rebuttal. Can you review it and confirm if there are any further concerns about the work?
>
> AC

---

### Meta-Review · Area_Chair_AiBs · 2026-01-04

**Summary:**

The paper addresses the "one-epoch overfitting" phenomenon prevalent in large-scale CTR/CVR prediction models. The authors provide a theoretical analysis based on Rademacher complexity to attribute this issue to the unconstrained growth of embedding norms for sparse, infrequent features. Based on this, they propose an adaptive regularization method that adjusts the regularization strength based on the update interval of features.

Reviewers initially appreciated the practical relevance, the theoretical motivation, and the simplicity of the proposed solution. However, concerns were raised regarding:
1.  **Theoretical Assumptions & Rigor:** Reviewer **amMd** questioned the validity of the i.i.d. assumption in industrial settings, the impact of stochastic noise in interval estimation, and the smoothness assumptions used in the derivations.
2.  **Baselines & Experimental Scope:** Reviewer **oGeC** criticized the experiments for relying on outdated backbones and requested comparisons with SOTA models like xDeepFM or DCNv2. Reviewer **amMd** requested comparisons with modern adaptive optimizers like SAM.
3.  **Model Scope & Analysis:** Reviewer **oGeC** asked to extend the complexity analysis to explicit interaction models (Factorization Machines) and investigated the connection to "dimensional collapse." Reviewer **fK43** noted a lack of comparison with LLM-based recommendation models.
4.  **Mechanisms & Ablations:** Reviewers **s7gA** and **fK43** requested fine-grained "bucket analyses" (performance by feature frequency) to verify that the regularization specifically targets tail features, and **fK43** further asked for transparency regarding hyperparameter search and computational overhead.

The authors provided a comprehensive rebuttal, adding new experiments, theoretical extensions, and analytical insights. Reviewer **oGeC** explicitly confirmed that their concerns were resolved and they would raise their score. The other reviewers' concerns were also well-addressed by the additional ablations and clarifications. Consequently, the paper is recommended for acceptance.

**Reviewer Concerns:**

**Addressed Concerns:**
*   **Baselines and Architecture (Reviewers oGeC, amMd):** The authors added experiments using xDeepFM and DCNv2 backbones (addressing **oGeC**), and compared against SAM (addressing **amMd**), demonstrating consistent improvements.
*   **Theoretical Validity (Reviewers amMd, oGeC):** The authors clarified the i.i.d. assumption in the context of incremental training, used Clarke subdifferential for non-smooth analysis (addressing **amMd**), and extended the Rademacher complexity analysis to Factorization Machines (addressing **oGeC**).
*   **Dimensional Collapse (Reviewer oGeC):** The authors provided a singular spectrum analysis showing their method preserves information abundance better than baselines.
*   **Mechanism Verification (Reviewers s7gA, fK43):** The authors provided the requested bucket analysis, showing that low-frequency features are indeed heavily regularized while high-frequency ones are preserved.
*   **Transparency & Complexity (Reviewer fK43):** Grid search details and memory complexity analysis (shared states for embedding rows) were provided.

**Outstanding Concerns:**
*   **LLM-based RecSys Comparison (Reviewer fK43):** The reviewer asked for comparisons to LLM-based recommenders. The authors argued that LLMs operate in a different parameter regime (dense-heavy) compared to the sparse-embedding-heavy models studied here. While no experiments were added, this justification is acceptable given the specific scope of the paper.

**Reviewer Scores:**

*   **Reviewer amMd:** 6 -> 6. The reviewer initially rated the paper as "marginally above acceptance" (6), appreciating the soundness but noting concerns about theoretical assumptions and baselines. While the rebuttal addressed the specific questions (e.g., SAM comparison, noise analysis), the reviewer's overall assessment of the paper's significance and the gap between theory and industrial reality likely remains unchanged. The score stays at 6, indicating the rebuttal solidified the acceptance case without elevating the paper to a "good" or "strong" accept category.

*   **Reviewer s7gA:** 8 -> 8. This reviewer was highly positive from the start ("Accept", 8). The rebuttal effectively confirmed the reviewer's intuition regarding the method's connection to MEDA and provided the requested bucket analysis. These responses reinforced the reviewer's initial high opinion, validating the score of 8 without introducing new factors to change it.

*   **Reviewer fK43:** 6 -> 6. The reviewer initially gave a 6, raising issues regarding hyperparameter transparency, ablations, and a lack of comparison to LLM-based models. Although the authors addressed the transparency and ablation requests, they argued that LLM-based comparisons were out of scope. The reviewer would likely appreciate the clarifications but maintain the score at 6, viewing the paper as a solid, practical contribution that meets the standard but does not exceed it due to the limited scope or novelty relative to broader trends.

*   **Reviewer oGeC:** 4 -> 6. This reviewer initially recommended rejection (4) due to the use of "outdated" experimental backbones. The authors directly addressed this primary blocker by adding experiments with SOTA models (xDeepFM, DCNv2) and providing the requested spectral analysis. The reviewer explicitly stated in the discussion that their concerns were resolved and they would raise their score. A shift to 6 reflects that the paper is now technically sound and acceptable, having fixed the major flaws that previously justified rejection.

---

### Decision · Program_Chairs · 2026-01-26

Accept (Poster)